# DND: Boosting Large Language Models with Dynamic Nested Depth

**Tieyuan Chen**[1,2,3*]**, Xiaodong Chen**[2,4]**, Haoxing Chen**[2]**, Zhenzhong Lan**[2,5]**,
Weiyao Lin**[1,3†]**, Jianguo Li**[2†]

[1]Shanghai Jiao Tong University,    [2]Inclusion AI, Ant Group    [3]ZhongguanCun Academy
[4]Renmin University of China,    [5]Westlake University

`{tieyuanchen, wylin}@sjtu.edu.cn, lijg.zero@antgroup.com`

## Abstract

We introduce Dynamic Nested Depth (DND), a novel method that improves performance for off-the-shelf LLMs by selecting critical tokens to reprocess in a nested depth manner. Specifically, at the end of the given transformer layer, DND identifies more critical tokens with a router and feeds them back for an extra round of processing, effectively "reviewing" difficult tokens while avoiding redundant computation for easier ones. The dynamic selection mechanism is tailored for precise control via two novel strategies: a router controlling loss to enhance token selection distinguishability, and a threshold control scheme to ensure selection stability. We demonstrate the effectiveness of DND by directly integrating it into pretrained dense and MoE models during a post-training phase. On diverse benchmarks, DND boosts the performances of the dense Qwen3-1.7B, Llama3.2-1B, and Gemma3-1B by 1.88%, 2.61%, and 2.50% and the MoE Qwen3-30B-A3B by 0.87%, all with a minimal parameter and computing increase.

## 1  Introduction

Large Language Models (LLMs) have transformed artificial intelligence with their powerful abilities. The main strategy for improving them has been scaling, as empirical laws show that model performance predictably increases with more parameters, data, and computation (Achim et al., 2023; Team et al., 2024; Yang et al., 2025; Liu et al., 2024). However, this scaling paradigm has significant drawbacks. The computational overhead for both training and inference grows exponentially with model size. This trend underscores a critical need for more efficient approaches to enhance model performance beyond simple brute-force scaling.

A key insight from (Gloeckle et al., 2024) is that prediction difficulty varies significantly across tokens; some are trivial to predict, while others demand deep computational processing. This disparity motivates token-level adaptive computation, where models can focus resources on the most critical inputs. A foundational version of this approach is token pruning, which has been shown to be effective across language understanding (Bae et al., 2025), model compression (Yang et al., 2022b), and vision (Hojjat et al., 2025). By filtering out uninformative tokens, these methods reduce computational overhead and can even improve robustness by mitigating noise. This establishes a binary choice: a token is either discarded or processed normally. Furthermore, we propose the natural next step: instead of merely retaining challenging tokens for standard processing, we should allocate additional computation to them, ensuring these critical tokens are properly understood.

Our choice of how to allocate additional computation is inspired by latent strategies in test-time scaling (Hao et al., 2024; Saunshi et al., 2025). Unlike using explicit output expansion like COT, these methods recur computation in hidden states, scaling inference without extra text generation. They observed that reasoning tokens place uneven demands on computation: as illustrated in Fig. 1, most tokens serve fluency, while a few critical ones drive complex planning or logical transitions.

Inspired by these two perspectives, we propose to integrate token-level selection with latent-space deepening. Instead of uniformly applying extra recurrent depth to all tokens, we dynamically select

---

\* This work was done when Tieyuan Chen was a research intern at Ant Group.
† Corresponding authors.

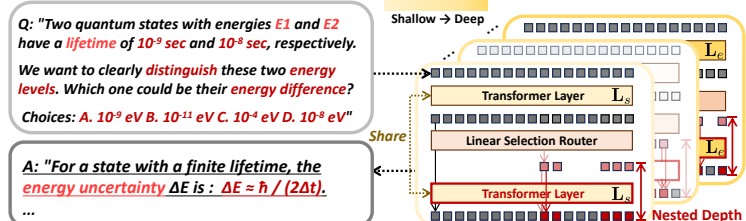

Figure 1: **DND Motivation.** The tokens highlighted in red denote critical elements in the QA pair. We propose a strategy within the transformer layers to identify and allocate additional computation to these critical tokens. $\mathbf{L}_s$ and $\mathbf{L}_e$ indicate the starting and end layers that adopted this strategy.

the subset of tokens that pose greater difficulty and reprocess them through transformer layers. This design not only concentrates additional computation on the most critical tokens but also allows the model to refine their hidden representations through internal "review" iterations.

To achieve dynamic selection and recalculation of critical tokens, we propose a novel architecture and training strategy. Specifically, as shown on the right of Fig. 1, we incorporate a linear layer as a router at the end of the transformer block with DND strategies. To achieve more robust routing and avoid potential information leakage during the inference of auto-regressive LLMs (Raposo et al., 2024), we adopt a token-choice routing strategy. In this approach, each token is routed independently based on whether its output exceeds a predefined threshold. The selected tokens are reorganized into a new sequence and are re-fed to compute the dynamic nested output. Moreover, as DND is a post-training method, we carefully design a normalized fusion strategy that integrates the dynamic nested output with the original forward output to preserve global pre-training knowledge.

In terms of training strategies, since our routing method treats each token independently, it lacks the precision of the top-k ratio routing when selecting tokens for recurrent computation (Raposo et al., 2024). To address this, we carefully design training strategies to control both the routers that determine token selection preferences and the thresholds that ultimately decide whether a token is selected. To enhance token selection distinguishability, we optimize the router's output distribution, encouraging the outputs across tokens to be distinguishable through a router controlling loss. To stabilize the token selection ratio during training, we adopt a threshold control scheme, where the threshold is updated based on the error between the expected ratio and the actual ratio computed over a sample buffer. Furthermore, to ensure more synchronized control, we update the threshold by blending it with the average top-k routing values using an exponential moving average (EMA).

Our method is a post-training approach that can be directly integrated into existing dense and Mixture-of-Experts (MoE) architectures. Experiments demonstrate its efficacy: by effectively selecting tokens, it substantially improves model performance across language, mathematics, reasoning, and coding tasks. The effectiveness is validated on both three small-scale dense models, Qwen3-1.7B, Llama3.2-1B, Gemma3-1B and a large-scale sparse MoE model, Qwen3-30B-A3B. Overall, our core contributions are:

- We introduce Dynamic Nested Depth (DND), an efficient paradigm that adaptively identifies critical tokens and selectively deepens their computation via nested re-processing.
- We design a tailored training strategy with a routing distribution control for token selection precision and an adaptive threshold control scheme for selection stability.
- Extensive experiments show that DND can be directly integrated into both dense and MoE architectures through post-training to achieve notable performance gains with minimal parameter and computation increase.

## 2 RELATED WORKS

### 2.1 ADAPTIVE TOKEN SELECTION

Token-level adaptive selection is most commonly applied in model quantization and compression (Yang et al., 2022b;a; Bondarenko et al., 2021), where Bayesian optimization methods are used to determine the appropriate compression ratios for tokens with varying levels of importance. This approach not only reduces computational redundancy but also mitigates the adverse effects of irrelevant tokens on the model's attention mechanism. Beyond the realm of model compression, token selection has also been explored in the field of computer vision (Luo et al., 2025; Gadhikar et al.,

2024; Hojjat et al., 2025). Motivated by the fact that visual representations often contain significant redundancy—due to irrelevant background information and high similarity between neighboring tokens—researchers have developed adaptive token selection strategies to address this issue. This strategy has been successfully applied to various vision tasks, including classification, detection, and retrieval. In general-purpose models, token selection is most notably employed in Mixture-of-Experts (MoE) architectures (Jiang et al., 2024; Yang et al., 2025; Liu et al., 2024), where a linear router dynamically assigns input tokens to specialized expert modules. Building on these insights, we propose the Dynamic Nested Depth (DND) approach, which adaptively selects tokens that are critical to represent and dynamically extends the model's depth for their processing.

## 2.2 Dynamic Reasoning Depth

Dynamic adjustment mechanisms for inference paths can generally be classified into two primary approaches. The first branch focuses on reducing computational redundancy, such as early exit (Zhou et al., 2025; Leviathan et al., 2023) and MOD (Raposo et al., 2024), which dynamically reduce the depth of computation layers to lower redundancy. The second branch investigates the test-time scaling law, which shows that repeatedly processing tokens within a single layer can enhance final inference accuracy. A sophisticated variant of this approach is the Latent Strategy (Hao et al., 2024; Saunshi et al., 2025), where reasoning is carried out within hidden states—either by completing all steps before generating an answer or by leveraging recurrent inference to iteratively refine them.

The most closely related studies to our work are ITT (Chen et al., 2025) and MOR (Bae et al., 2025), both of which dynamically select a subset of tokens for additional computation and yield certain performance improvements. While sharing MOR (Bae et al., 2025)'s goal of improving performance via dynamically increased computational depth, the two still differ fundamentally. MOR attempts to improve parameter efficiency during pretraining through a recurrent structure, which requires training a model from scratch on over 200B tokens. This is extremely costly and makes it difficult to apply the approach directly to existing open-source SOTA models. In contrast, DND focuses on unlocking the potential of existing state-of-the-art pretrained models and proposes a plug-and-play post-training method. Besides, our work differs from MOR in model scale, training phase, architecture, and routing control (detailed in Appendix. Sec . C). MOR is limited to 1B-parameter, whereas our DND successfully scales to a 30B MoE model. We also address a key limitation in token selection control. Unlike MOR, which relies on z-loss (Zoph et al., 2022) for approximate load balancing, we achieve precise, stable token selection. Our method jointly enhances routing discriminability and adjusts thresholds via EMA-synchronized buffer errors.

## 3 Methodology

Our method is primarily divided into two main parts: the model architecture design (Sec. 3.1) and the training strategies (Sec. 3.2), where we detail how we implement dynamic nested depth (DND) and the carefully designed training strategies used to ensure the effectiveness of the architecture.

### 3.1 Architecture

In the model architecture section, we will introduce how tokens are selected (Sec. 3.1.1), how the new nested depth output is computed (Sec. 3.1.2), and how the vanilla output and dynamic nested output are fused to obtain the final output (Sec. 3.1.3). The whole architecture is shown in Fig. 2. Moreover, we apply the DND strategy only to the intermediate layers of the model, keeping the initial and final layers unchanged to preserve the reasoning patterns learned during pre-training (Ma et al., 2023; Xia et al., 2023). The layers where the DND is applied are denoted as from $\mathbf{L}_s$ to $\mathbf{L}_e$.

### 3.1.1 Routing Design

When considering the routing paradigm, as shown in Fig. 3 (a), routing the entire sequence with expert choice creates a mismatch with the next-token prediction paradigm of auto-regressive models. This is because the full sequence cannot be accessed during early decoding without risking information leakage (Raposo et al., 2024). To address this, we adopt token-choice selection, as illustrated in Fig. 3 (b), which dynamically decides whether each token should undergo further processing.

Concretely, after an initial forward pass through a transformer layer, we obtain the hidden states of the token sequence, referred to as the vanilla output $\mathbf{X}_v$. To determine token preferences for further computation, we employ a router similar to that in MoE architectures, implemented as a simple

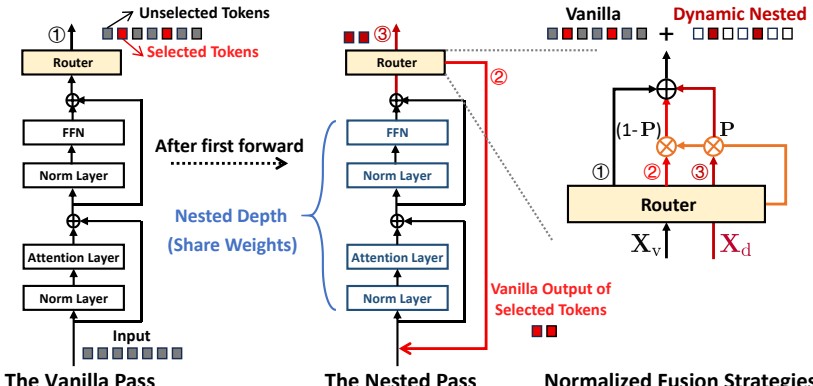

Figure 2: **DND Framework.** The central idea of DND is a dynamic nested pass of critical tokens after the vanilla forward process of the transformer layers. Whether a token is selected or not is determined by a router. The block's final output is a merged result of vanilla output and nested output, governed by normalized routing weights.

linear layer $R : \mathbb{R}^{d_{\text{model}}} \to \mathbb{R}$, where $d_{\text{model}}$ denotes the hidden size of the transformer model. For each token in the sequence, the router takes its hidden state from the vanilla pass, $\mathbf{x}_{\text{v}}^i \in \mathbb{R}^{d_{\text{model}}}$, as input and distributes a preference score. This score is then normalized using a sigmoid function, $\sigma(\cdot)$, to yield a probability $p^i \in (0, 1)$:

$$p^i = \sigma(R(\mathbf{x}_{\text{v}}^i)) \tag{1}$$

The selection decision for each token is made independently by comparing its routing probability $p^i$ against a pre-defined threshold $\tau$. A token $i$ is selected for reprocessing if and only if $p^i > \tau$.

### 3.1.2 NESTED DEPTH DESIGN

Once the tokens for recurrence are identified, they undergo a nested processing pass through the *same* transformer layer. We construct a binary mask $\mathbf{M}$ according to the routing result, where each element $m^i$ is defined as:

$$m^i = \begin{cases} 1, & \text{if } p^i > \tau \\ 0, & \text{if } p^i \leq \tau \end{cases} \tag{2}$$

With the binary mask, the chosen states are assembled into a compact sequence for recurrent computation, refining the representations of the selected tokens. This process can be expressed as:

$$\mathbf{X}_{\text{d}} = \text{Unpack}(\mathbf{L}_i(\text{Pack}(\mathbf{X}_{\text{v}}, \mathbf{M}) + \mathbf{E}'_{\text{pos}}), \mathbf{M}) \tag{3}$$

where $\mathbf{X}_{\text{d}}$ are the output hidden states from this dynamic nested pass. The $\text{Pack}(\mathbf{X}_{\text{v}}, \mathbf{M})$ operator selects tokens from the input sequence $\mathbf{X}_{\text{v}}$ using a mask $\mathbf{M}$ to form a compact subsequence. This subsequence is then given new positional embeddings $\mathbf{E}'_{\text{pos}}$ and processed by the $i$-th transformer layer $\mathbf{L}_i$. Finally, the Unpack operator scatters the results back to their original positions within a zero-padded tensor, guided by the same mask $\mathbf{M}$. This recurrence allows the model to perform internal "review" iterations, dedicating additional computational depth to refine the representations of the critical tokens without altering the simpler ones.

### 3.1.3 FUSION DESIGN

To ensure that the model effectively enhances the representations of critical tokens via the DND strategy while retaining the knowledge of global token interactions acquired during pretraining, we propose a normalized fusion strategy. Specifically, we merge the outputs from the vanilla pass ($\mathbf{X}_{\text{v}}$) and the dynamic nested pass ($\mathbf{X}_{\text{d}}$) using a gating mechanism. The final output $\mathbf{X}$ is computed as:

$$\mathbf{x}^i = \begin{cases} (\beta \cdot p^i) \cdot \mathbf{x}_{\text{d}}^i + (1 - \beta \cdot p^i) \cdot \mathbf{x}_{\text{v}}^i, & \text{if } p^i > \tau \\ \mathbf{x}_{\text{v}}^i, & \text{if } p^i \leq \tau \end{cases} \tag{4}$$

Here, $\mathbf{x}^i$ refers to the merged hidden state of the $i$-th token. $\beta$ is a learnable parameter that acts as a balancing factor between the original and nested paths. Similar to (Raposo et al., 2024), this

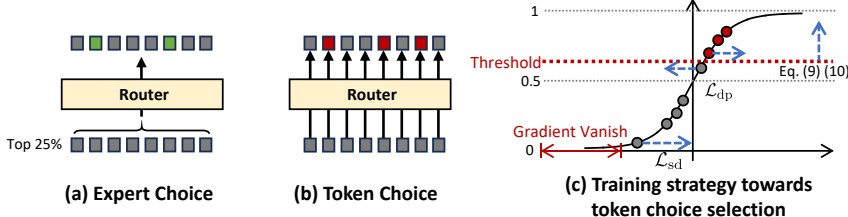

Figure 3: **Routing Design and Training Strategies.** Figure (a) illustrates expert-choice routing, where the top-k proportion is selected over the entire sequence. Figure (b) shows token-choice routing, which selects tokens independently and suits auto-regressive models. Figure (c) summarizes our training strategy: routing outputs are optimized to enhance token distinguishability by dispersing the token-level routing distribution via $\mathcal{L}_{\text{sd}}$ and preventing it from collapsing into gradient-vanishing regions via $\mathcal{L}_{\text{dp}}$. In addition, buffer proportional control (Eq. (9)) and EMA synchronization (Eq. (10)) effectively regulate the stability of the selection by computing the real-time error ratio.

fusion is modulated by the token's own routing score $p^i$, ensuring that tokens deemed more difficult (higher $p^i$) incorporate a larger portion of their recomputed representation. This design provides a smooth and adaptive integration, effectively stabilizing the learning process and allowing the model to control the influence of the additional computation dynamically.

## 3.2 TRAINING STRATEGIES

Our training strategy is primarily designed to ensure that the model successfully learns to distinguish tokens in order to perform recurrent computation. As our token-choice routing lacks the explicit ratio control of top-k mechanisms, we carefully designed strategies to control the two key factors in selection: the router's output distribution (Sec. 3.2.1) and the selection threshold (Sec. 3.2.2).

### 3.2.1 ROUTER CONTROLLING LOSS

A primary challenge in controlling the router is ensuring its output scores, $p^i$, are sufficiently distinguishable. If scores cluster within a narrow range, the token selection process becomes unstable, as minor fluctuations in the threshold $\tau$ can cause drastic changes in the selection ratio.

To address this, as shown in Fig. 3 (c), we introduce a control strategy built upon a dual-objective loss function. The core idea is to create a dynamic tension between two competing goals:

1. **Score Dispersion:** We encourage the scores within a sequence to spread out across a wide range. This makes the selection robust by creating clear distinctions between tokens.

2. **Distribution Preservation:** We simultaneously constrain the scores to remain near the center of the sigmoid function's dynamic range. This ensures the router remains sensitive and responsive to its inputs, avoiding gradient vanish.

These competing objectives work in concert to produce a distribution of routing scores that is both discriminative and stable. We formulate the final routing objective as the total router loss, $\mathcal{L}_{\text{router}}$, which is jointly optimized with the model's main cross-entropy loss. The loss is defined as:

$$\mathcal{L}_{\text{router}} = \lambda_{\text{sd}}\mathcal{L}_{\text{sd}} + \lambda_{\text{dp}}\mathcal{L}_{\text{dp}} \tag{5}$$

where $\mathcal{L}_{\text{sd}}$ is the *Score Dispersion Loss* and $\mathcal{L}_{\text{dp}}$ is the *Distribution Preservation Loss*. The hyperparameters $\lambda_{\text{sd}}$ and $\lambda_{\text{dp}}$ balance the influence of each component.

**Score Dispersion Loss.** To counteract the tendency for router scores to cluster, we apply a *Score Dispersion Loss*, $\mathcal{L}_{\text{sd}}$. This loss, based on information entropy, is designed to push the score distribution towards diversity at each targeted layer. For each layer $l$ in this range, we take its sequence of $N$ routing scores $\mathbf{p}^{(l)} = \{p^{1,(l)}, \ldots, p^{N,(l)}\}$ and normalize them to form a distribution: $p'^{,i,(l)} = p^{i,(l)}/\sum_{j=1}^{N} p^{j,(l)}$. The total loss is the sum of the information entropy from each layer, turning the goal of maximizing entropy into a minimization problem for the optimizer:

$$\mathcal{L}_{\text{sd}} = \sum_{l=\mathbf{L}_s}^{\mathbf{L}_e} \left(-H(\mathbf{p}'^{(l)})\right) = -\sum_{l=\mathbf{L}_s}^{\mathbf{L}_e} \sum_{i=1}^{N} p'^{,i,(l)} \log(p'^{,i,(l)}) \tag{6}$$

This formulation incentivizes the router to produce a diverse set of scores, making the routing output discriminative enough across tokens, therefore less sensitive to minor threshold adjustments.

**Distribution Preservation Loss.**   While the router loss, $\mathcal{L}_{\text{router}}$, promotes a dispersed distribution of routing scores, its reliance on a sigmoid activation function leads to vanishing gradients as outputs approach 0 or 1. This issue is particularly pronounced when the target selection ratio is low (e.g., 20%) or high (e.g., 80%), as the model may push many scores into the sigmoid's saturation regions. Consequently, the model may lose the ability to discriminate between tokens with scores near the decision threshold. To mitigate this, we introduce a *Distribution Preservation Loss*, $\mathcal{L}_{\text{dp}}$, which counteracts this effect by applying a Mean Squared Error penalty to scores that deviate from 0.5, thereby preserving gradient flow and enhancing discriminability:

$$\mathcal{L}_{\text{dp}} = \sum_{l=\mathbf{L}_s}^{\mathbf{L}_e} \left( \frac{1}{N} \sum_{i=1}^{N} (p^{i,(l)} - 0.5)^2 \right) \tag{7}$$

where $p^{i,(l)}$ is the score of the $i$-th token at layer $l$. This objective effectively pulls the score distributions towards the center of the sigmoid's dynamic range, ensuring the routers remain responsive to changes in token hidden states.

Together, these two losses create a balanced "push-pull" dynamic. The entropy-based dispersion loss pushes scores apart to cover a wider spectrum, while the MSE-based preservation loss pulls them collectively towards the responsive center. The result is a router that produces scores that are discriminative enough, facilitating more accurate and reliable token selection.

### 3.2.2   THRESHOLD CONTROL SCHEME

The preceding section introduced our method for enhancing the discriminability of router outputs. To further refine token selection, we propose a dynamic threshold control scheme, which adaptively regulates the threshold to achieve the desired proportion of selected tokens flexibly. Previous approaches using z-loss could only balance between selecting and not selecting tokens (Bae et al., 2025). Inspired by the balance loss proposed in DeepSeek-V3 (Liu et al., 2024), we design a method that computes the average selection ratio error in a buffer to adjust the threshold accordingly. Additionally, we leverage an EMA synchronization to assist in optimizing the threshold.

**Buffer Proportional Control.**   Our primary mechanism for threshold control is a loss-free method that makes real-time adjustments to the threshold $\tau$. For each mini-batch $\mathcal{B}$ of training steps, this controller computes an error signal, $e$, representing the deviation between the actual selection ratio during training and a pre-defined target ratio, $k_{\text{target}}$. $e$ is formulated as:

$$e = \frac{\sum_{b \in \mathcal{B}} \sum_{i=1}^{N_b} m_b^i}{\sum_{b \in \mathcal{B}} N_b} - k_{\text{target}} \tag{8}$$

where $N_b$ is the number of tokens in sample $b$, and $m_b^i$ is the binary selection mask. Based on this error, the threshold is immediately updated via a simple but effective control law:

$$\tau \leftarrow \tau + \alpha \cdot e \tag{9}$$

where $\alpha$ is a small step size (proportional gain). This mechanism provides immediate feedback to stabilize the selection ratio against short-term fluctuations. If too many tokens are selected ($e > 0$), $\tau$ increases to induce a decreasing selection trend; if too few are selected ($e < 0$), $\tau$ decreases.

**EMA Synchronization.**   While the buffer proportional controller excels at rapid, local adjustments, its effectiveness can degrade when the optimization directions of routing and the threshold are misaligned. To prevent this drawback, we introduce an auxiliary mechanism that acts as a low-frequency synchronization loop. Periodically (e.g., every 50 steps), we compute a smoothed ideal threshold. Specifically, we maintain a buffer of the most recent $N_b$ samples. For each step in this buffer, we calculate its corresponding $\tau_{\text{topk}}$—the threshold value that would have precisely selected the target ratio. The average of these values, denoted as $\bar{\tau}_{\text{topk}}$, serves as a more stable and robust estimate of the ideal threshold around these optimization steps. The operational threshold $\tau$ is then gently nudged towards this averaged estimate using an exponential moving average (EMA):

$$\tau = (1 - \gamma) \cdot \tau + \gamma \cdot \bar{\tau}_{\text{topk}} \tag{10}$$

where $\gamma$ is a smoothing factor. This process ensures the router and threshold remain synchronized, preventing sustained periods of over- or under-selection and promoting long-term training stability.

Table 1: **SFT Performance Comparison of Different Small-Scale Dense LLMs.** Performing full-scale SFT with the DND strategy on the three widely used base models yields additional average improvements of 1.88, 2.61, and 2.50 points over full-scale SFT alone.

| | Average | General Knowledge & Alignment | | | | | Math & STEM | | Coding & Agent | | | |
| | | BBH | PIQA | C-Eval | MMLU | IFEval | GPQA | GSM8K | MBPP | Human | BFCL | MultiPLE |
|---|---|---|---|---|---|---|---|---|---|---|---|---|
| Qwen3-1.7B | 59.53 | 40.82 | 75.38 | 60.00 | 64.11 | 65.47 | 28.54 | 79.38 | 68.38 | 61.59 | 58.73 | 52.45 |
| + ITT | 59.58 | 41.23 | 75.99 | 59.51 | 64.92 | 65.08 | 27.85 | 80.29 | 68.67 | 60.78 | 58.62 | 52.45 |
| **+ DND** | **61.41** | **45.84** | **76.25** | **60.38** | **64.45** | **66.87** | **34.34** | **80.15** | **71.90** | **62.71** | **59.80** | **52.80** |
| $\Delta$ *(+-)* | *+1.88* | *+5.02* | *+0.87* | *+0.38* | *+0.34* | *+1.40* | *+5.80* | *+0.77* | *+3.52* | *+1.12* | *+1.07* | *+0.35* |
| Llama3.2-1B | 45.37 | 25.73 | 65.48 | 47.82 | 53.28 | 52.45 | 10.73 | 63.23 | 49.54 | 50.42 | 44.89 | 35.47 |
| **+ DND** | **47.98** | **29.43** | **66.51** | **49.21** | **55.63** | **55.68** | **14.59** | **66.57** | **52.91** | **52.16** | **47.52** | **37.56** |
| $\Delta$ *(+-)* | *+2.61* | *+3.70* | *+1.03* | *+1.39* | *+2.35* | *+3.23* | *+3.86* | *+3.34* | *+3.37* | *+1.74* | *+2.63* | *+2.09* |
| Gemma3-1B | 47.08 | 25.93 | 70.27 | 50.14 | 55.98 | 54.52 | 16.49 | 65.53 | 49.29 | 52.62 | 40.69 | 36.43 |
| **+ DND** | **49.58** | **30.62** | **71.33** | **51.00** | **58.04** | **56.91** | **21.79** | **68.68** | **52.96** | **53.76** | **42.88** | **37.41** |
| $\Delta$ *(+-)* | *+2.50* | *+4.69* | *+1.06* | *+0.86* | *+2.06* | *+2.39* | *+5.30* | *+3.15* | *+3.67* | *+1.14* | *+2.19* | *+0.98* |

## 4 EXPERIMENTS

### 4.1 EVALUATION BENCHMARK

We provide an extensive empirical evaluation of DND over a wide variety of benchmarks, demonstrating its effectiveness and robustness. The evaluation suite covers three primary domains: **1. General Knowledge & Alignment:** MMLU (Hendrycks et al., 2020), CEval (Huang et al., 2023), CMMLU (Li et al., 2023), BBH (Srivastava et al., 2022), DROP (Dua et al., 2019), IFEval (Zhou et al., 2023), PIQA (Bisk et al., 2020). **2. Mathematics & STEM:** Math (Hendrycks et al., 2021), GSM8k (Cobbe et al., 2021), MATH-500, AIME24, GPQA-Diamond (Rein et al., 2023),. **3. Coding & Agent:** MBPP, MBPP+ (Austin et al., 2021), HumanEval+ (Chen et al., 2021), LCB-v5(LiveCodeBench-v5) (Jain et al., 2024), LCB-v6(LiveCodeBench-v6) (Jain et al., 2024), MultiPL-E (Cassano et al., 2022), BFCL v3 (Live).

### 4.2 TRAINING DETAILS

Our DND model undergoes standard full-scale supervised fine-tuning (SFT) using a comprehensive and diverse dataset, with all parameters set as trainable and the same learning rate applied. Our training data incorporates a significant volume of synthetic material built upon a high-quality seed set of 1-2 million instances curated from human annotations and open-source materials. The model's weights are initialized from the Qwen3-1.7B Base, Llama-3.2-1B, Gemma3-1B-pt, and Qwen3-30B-A3B Base. Detailed hyperparameters and training settings are provided in Appendix Sec.B.

### 4.3 MAIN RESULTS

**Base Evaluation.** As shown in Tab. 1, our method achieves obvious improvements across the three widely used base models. Especially on datasets that require complex reasoning, such as BBH and GPQA, the performance boost is particularly notable, with all three models showing an additional performance improvement of around 5%. Additionally, we found that when the SFT is conducted with ITT (Chen et al., 2025) under the same computation cost, the performance improvement is not as pronounced. The limited performance gains stem from the use of Top-P–based token selection for auto-regressive LLM, which introduces a mismatch between training and inference, and may also lead to potential information leakage according to (Raposo et al., 2024).

**Scaling Evaluation.** As shown in Tab. 2, our DND strategy consistently improves the performance of the Qwen3-30B-A3B model, achieving an average gain of +0.87 across 17 benchmarks without any performance degradation. The impact of DND is most pronounced in Coding and Agent tasks, yielding notable gains of +2.05 on BFCL v3, +1.42 on LCB-v6, and +1.24 on LCB-v5. These results strongly support our hypothesis that DND effectively filters extraneous noise, allowing the model to focus its capacity on sparse, high-value tokens essential for complex reasoning, planning, and code generation. Importantly, the benefits are not limited to specialized domains: substantial improvements are also observed in General and Alignment tasks (+1.83 on C-Eval), alongside robust generalization on challenging Math and STEM benchmarks. Crucially, these substantial performance gains are realized with a negligible increase of only 0.03M parameters, highlighting DND as a highly parameter-efficient approach for unlocking significant capabilities in LLMs. **FLOPs and Throughput Evaluation.** As demonstrated in Appendix Sec. A, reviewing 20% of the tokens adds

Table 2: **Performance Comparison of Qwen3-30B-A3B with and wothout DND.** The final column shows the difference ($\Delta$) between the SFT results of vanilla Qwen3-A3B-30B and Qwen3-A3B-30B+DND. And we list Qwen3-32B and Qwen3-30B-A3B Chat model for reference.

| Task | Qwen3-32B (Non-Thinking) | Qwen3-30B-A3B (Non-Thinking) | Qwen3-A3B-30B (SFT) | Qwen3-A3B-30B+DND (SFT) | $\Delta$ (w vs w/o DND) |
|---|---|---|---|---|---|
| **General & Alignment Tasks** | | | | | |
| MMLU | 82.93 | 80.12 | 85.41 | **85.91** | *+0.50* |
| CMMLU | 84.63 | 83.13 | 84.82 | **85.19** | *+0.37* |
| BBH | 85.45 | 82.55 | 86.90 | **87.03** | *+0.13* |
| DROP | 84.02 | 86.38 | 86.21 | **86.48** | *+0.27* |
| C-Eval | **87.53** | 85.95 | 83.09 | 84.92 | *+1.83* |
| IFEval | **85.27** | 84.55 | 83.09 | 84.31 | *+1.22* |
| **Mathematic & STEM Tasks** | | | | | |
| MATH | 85.26 | 84.68 | 88.63 | **88.78** | *+0.15* |
| MATH-500 | 87.40 | 88.70 | 92.60 | **92.80** | *+0.20* |
| GSM8K | 94.54 | **95.30** | 94.30 | 95.10 | *+0.80* |
| AIME24 | 27.71 | 28.33 | 51.46 | **52.37** | *+0.91* |
| GPQA-Diamond | 53.60 | 51.71 | 56.76 | **57.67** | *+0.91* |
| **Coding & Agent Tasks** | | | | | |
| HumanEval+ | 82.93 | 84.15 | 85.59 | **86.58** | *+0.99* |
| MBPP+ | 72.75 | 75.16 | 78.84 | **79.54** | *+0.70* |
| MultiPLE | 68.62 | 66.04 | 72.60 | **73.72** | *+1.12* |
| LiveCodeBench v5 | **31.44** | 28.89 | 29.94 | 31.18 | *+1.24* |
| LiveCodeBench v6 | 28.57 | 29.43 | 31.14 | **32.56** | *+1.42* |
| BFCL v3 (Live) | 75.09 | 73.69 | 75.43 | **77.48** | *+2.05* |
| **Average** | 72.81 | 73.44 | 75.70 | **76.57** | *+0.87* |

Table 3: **Comparison of Speed across Different Input and Decode Lengths.** The speeds are measured using BF16 Quantization of LLM and accelerated with vLLM kernels.

| Input Length | Decode Length | Speed (tokens/s) | | Relative Speed (%) |
|---|---|---|---|---|
| | | Qwen3-30B-A3B | + DND | |
| 1024 | 2048 | 148.68 | 136.19 | 91.6 |
| 6144 | 2048 | 208.60 | 193.51 | 92.8 |
| 1024 | 6144 | 76.16 | 70.52 | 92.6 |
| 6144 | 6144 | 100.89 | 93.93 | 93.1 |

only about 6% extra FLOPs when applying the DND strategy to the Qwen3-30B-A3B model. To further assess the inference speed of our model in practical settings relative to the baseline, following SGLang, as shown in the Tab. 3, we measured the throughput of both Qwen3-30B-A3B and Qwen3-30B-A3B+DND models under four standard sequence lengths using a single H100 GPU with a single batch. The measurement results in the table show that, while achieving performance improvements, our model consistently reaches 91.6-93.1% of the speed of the vanilla model under different circumstances.

## 4.4 ABLATION STUDY

As shown in Tab. 4, we conducted ablation experiments of the DND strategy on Qwen3-1.7B.

**Training Strategies.** We conducted ablation experiments on the proposed training strategy. We found that when using only the DND framework with a simple z-loss-like method to control token selection, performance dropped noticeably, yielding an average improvement of only 1.01 points over Qwen3-1.7B's SFT performance. This highlights the importance of our carefully designed training strategy. Moreover, router and threshold control function as complementary components for token selection control. While each method individually provides marginal gains, their combination leads to a clear improvement of approximately one percentage point in average accuracy.

**Hyper-parameters of Architecture.** Besides, we conducted ablation experiments on several important hyperparameters of the model architecture. For the expected token selection ratio, we tested 10%, 20%, and 30%. We found that when only 10% of tokens were selected, the number of tokens participating in attention computation was likely too small, resulting in a modest improvement of just 0.8% over the baseline. In contrast, selecting approximately 20–30% of tokens achieved better performance. To balance computational efficiency, we chose an expected selection ratio of 20% for our DND method in the Qwen3-30B-A3B scaling experiments. We also performed ablations on the number of shallowest and deepest layers retained in the original architecture, finding that keeping about four layers at both the beginning and the end yielded the best performance. This configuration was retained in the DND experiments on Qwen3-30B-A3B.

Table 4: **Ablation study of Qwen3-1.7B with DND under different settings.** TC indicates threshold control (including buffer proportional control and EMA synchronization), RC indicates router control (including proposed $\mathcal{L}_{sd}$ and $\mathcal{L}_{dp}$), layer ($\mathbf{L}_s : \mathbf{L}_e$) indicates the layers with DND.

| Metric / Setting | Qwen3-1.7B | +DND | +DND | +DND | +DND | +DND | +DND | +DND | +DND |
|---|---|---|---|---|---|---|---|---|---|
| RC ($\mathcal{L}_{sd}$ and $\mathcal{L}_{dp}$) | – | ✓ | × | × | ✓ | ✓ | ✓ | ✓ | ✓ |
| TC (BPC & EMA) | – | ✓ | × | ✓ | × | ✓ | ✓ | ✓ | ✓ |
| $k_{target}$(%) | – | 20 | 20 | 20 | 20 | 10 | 30 | 20 | 20 |
| layer ($\mathbf{L}_s : \mathbf{L}_e$) | – | 4: 23 | 4: 23 | 4: 23 | 4: 23 | 4: 23 | 4: 23 | 5: 22 | 3: 24 |
| Average | 59.53 | **61.41** | 60.54 | 60.58 | 60.68 | 60.33 | 61.03 | 61.05 | 60.36 |
| Δ (+-) | 0.00 | **1.88** | 1.01 | 1.05 | 1.15 | 0.80 | 1.50 | 1.52 | 0.83 |
| BBH | 40.82 | **45.84** | 44.69 | 43.47 | 44.37 | 44.95 | 45.17 | 44.85 | 44.95 |
| C-Eval | 60.00 | 60.38 | 60.21 | 60.38 | 60.17 | 59.61 | 60.32 | **60.44** | 59.61 |
| MMLU | 64.11 | 64.45 | 64.33 | 64.45 | 64.26 | 64.38 | **64.66** | 64.25 | 64.56 |
| GPQA-D | 28.54 | **34.34** | 29.92 | 31.94 | 32.79 | 29.17 | 33.82 | 33.82 | 29.26 |
| PIQA | 75.38 | 76.25 | 75.87 | 76.38 | 76.05 | 75.95 | **76.33** | 76.15 | 75.95 |
| BFCL | 58.73 | 59.80 | 59.21 | 59.32 | 59.50 | 59.40 | **59.92** | 59.60 | 59.40 |
| IFEval | 65.47 | 66.87 | 66.55 | 66.35 | 66.19 | 65.99 | 67.27 | **67.31** | 65.99 |
| GSM8K | 79.38 | **80.15** | 79.92 | 79.80 | 79.77 | 79.77 | 79.15 | 79.53 | 79.85 |
| Humaneval+ | 61.59 | 62.71 | 61.59 | 61.59 | 61.82 | 62.15 | 62.80 | **62.96** | 62.15 |
| MBPP | 68.38 | **71.90** | 70.73 | 69.84 | 69.79 | 69.32 | 69.09 | 69.56 | 69.32 |
| MultiPLE | 52.45 | 52.80 | 52.95 | 52.88 | 52.80 | 52.88 | 52.82 | **53.05** | 52.88 |

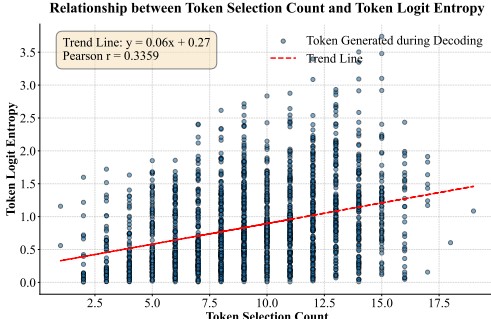

Figure 4a: **Entropy of Selected Tokens.** The selection frequency of a token correlates positively with its original logit entropy in the vanilla pass.

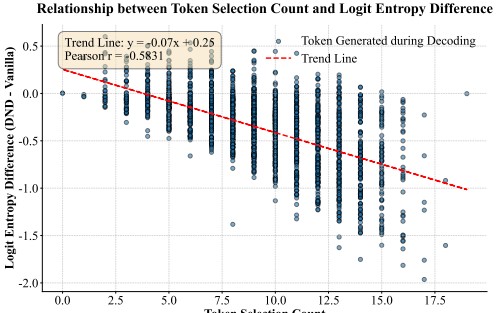

Figure 4b: **Logits Variation after DND.** For rarely selected tokens, logit entropy fluctuates evenly, but it decreases as selection frequency increases.

## 4.5 TOKEN SELECTION ANALYSIS

**Why critical tokens are selected?** As shown in Fig. 4a, to examine whether the tokens selected by our DND model are indeed critical, we analyze the relation between the selection frequency of tokens routed by Qwen3-30B-A3B+DND and the vanilla model's logit entropy without any reprocessing. According to the findings in (Ma et al., 2025), high entropy in the logit indicates that the model is uncertain about which vocabulary to select or is hesitating between multiple possible answers. We find that tokens with higher logit entropy are frequently selected by the router across multiple layers. This shows that DND preferentially selects tokens with greater uncertainty, validating the motivation behind critical token selection and confirming the effectiveness of the router.

**Why better representations are learned?** Furthermore, as shown in Fig. 4b, to validate that our DND strategy reduces the model's hesitation or uncertainty about critical tokens, we evaluated the variation in logit entropy for the same token after applying the DND strategy, compared to the entropy in the vanilla model. We found that after nested reviews, the logit entropy of the selected tokens significantly decreased, proving the effectiveness of our method.

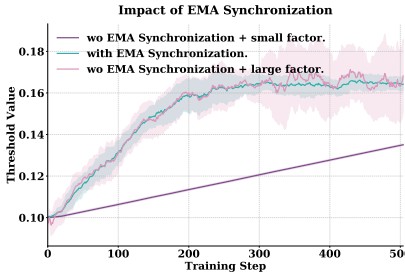

Figure 5: **Threshold Adjustment during Training.** With EMA synchronization, the threshold can be adjusted smoothly and in real time.

**Threshold Visualization during Training.** As illustrated in Fig. 5, we analyze the threshold dynamics of the 24th layer in our DND (Qwen3-30B-A3B) model by presenting the effect of two different approaches for threshold control. Adopting buffer proportional control alone exhibits a critical tuning challenge: a small adjustment factor (purple line) causes the threshold to adapt too slowly, consistently failing to reach the target selection ratio and thereby impairing early training performance. Conversely, an excessively large factor (pink line) leads to volatile oscillations around the target, which compromises stability. In contrast, incorporating our EMA synchronization (blue line) enables rapid threshold adjustments, maintaining synchronization between the router and threshold, thereby ensuring stable selection.

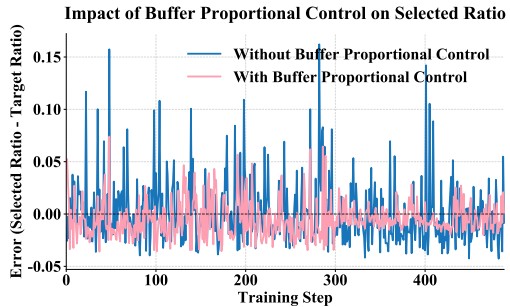

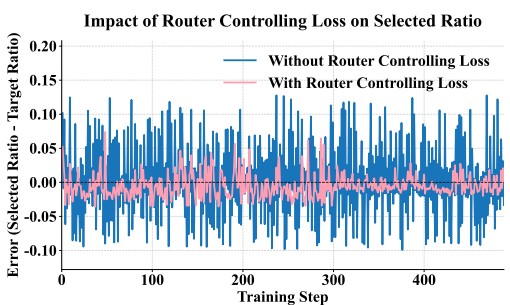

Figure 6a: **Selected Ratio Comparisons.** The average selection ratio is stably controlled with our proposed buffer proportional control strategy.

Figure 6b: **Selected Ratio Comparisons.** Oscillations arising from insufficient discrimination are reduced by introducing the router controlling loss.

**Selected Ratio Visualization during Training.** As shown in Fig. 6a, introducing EMA synchronization alone struggles to regulate token selection, as its non-real-time adjustments often misalign with the model's optimization trajectory, resulting in large and persistent oscillations. By introducing buffer proportional control, we provide a high-frequency corrective loop. As a result, oscillations are rapidly suppressed to within a tight 5% band, successfully stabilizing the selection. Besides, as shown in Fig. 6b, introducing the router controlling loss significantly reduces the magnitude of oscillations, demonstrating its effectiveness in enhancing the discriminability of token routing outputs.

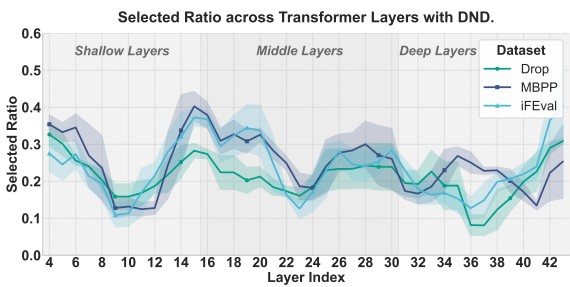

Q: "Two quantum states with energies E1 and E2 have a lifetime of 10^-9 sec and 10^-8 sec, respectively. We want to clearly distinguish these two energy levels. Which one could be their energy difference?
Choices: A. 10^-9 eV B. 10^-11 eV C. 10^-4 eV D. 10^-8 eV"

A: "For a state with a finite lifetime, the energy uncertainty ΔE is approximately: $\Delta E \approx \hbar / (2\Delta t)$.

For state 1: lifetime $\Delta t_1 = 10^{-9}$ s $\Delta E_1 \approx \hbar / (2\Delta t_1) = (6.582 \times 10^{-16}$ eV·s$) / (2 \times 10^{-9}$ s$) \approx 3.29 \times 10^{-6}$ eV

For state 2: lifetime $\Delta t_2 = 10^{-8}$ s $\Delta E_2 \approx \hbar / (2\Delta t_2) = (6.582 \times 10^{-16}$ eV·s$) / (2 \times 10^{-8}$ s$) \approx 3.29 \times 10^{-7}$ eV

To resolve two energy levels, the energy difference ΔE between them should be at least comparable to the larger of the two uncertainties. So only option C ($10^{-4}$ eV) is large enough to be resolved. Answer: C"

Figure 7a: **Selected Ratio during Evaluation.** Token selection ratio tends to be slightly higher in the middle layers as well as in the shallowest and deepest layers.

Figure 7b: **Visualization Results.** Highlighted tokens are frequently selected, with darker red indicating higher selection in deeper layers.

**Selected Ratio Visualization during Evaluation.** To validate our training methodology, we evaluated the DND model's inference-time behavior across a diverse suite of benchmarks, including agent tasks, reasoning, and code generation. The average token selection ratios across all layers consistently aligned with our target, ranging from 0.178 to 0.242, which confirms the effectiveness of our whole control strategy. A more granular, layer-wise analysis in Fig. 7a reveals a nuanced pattern where selection is slightly elevated in the middle layers and the first layers from each end. Complementing this quantitative success, a qualitative visualization of the GPQA dataset in Fig. 7b reveals an interesting phenomenon. Tokens selected by shallower layers (lighter colors) are predominantly essential nouns, while those selected by deeper layers (darker colors) correspond to more abstract or syntactically critical components like mathematical expressions and key verbs. This suggests that the model learns a hierarchical processing strategy, using earlier layers to identify key entities and later layers to perform more complex relational and logical operations.

## 5 CONCLUSION

In this work, we introduce Dynamic Nested Depth (DND), a novel and efficient method for enhancing Large Language Model performance. DND adaptively identifies critical tokens and selectively deepens their computation via nested re-processing. This is achieved through a token-choice routing design with a normalized output fusion strategy. The precision and stability of this selection process are guaranteed by our router controlling loss and threshold control scheme. We validated DND on both dense (Qwen3-1.7B, Llama3.2-1B, Gemma3-1B) and larger-scale sparse MoE (Qwen3-30B-A3B) models, demonstrating substantial accuracy improvements with a negligible parameter increase ($< 0.1$M) and minimal computing increase. These results affirm that targeted, dynamic nested depth computation is a powerful method for boosting LLMs' performance.

ACKNOWLEDGEMENT

The paper is supported in part by the National Natural Science Foundation of China (No.62325109, 62595733, 62561160155), the Shanghai 'The Belt and Road' Young Scholar Exchange Grant (24510742000), and in part by the Ant Group Research Intern Program.

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

# A COMPUTATIONAL OVERHEAD ANALYSIS OF THE DYNAMIC TOKEN RECALCULATION STRATEGY

We conduct a formal analysis of the computational overhead introduced by the dynamic nested depth strategy within a Qwen3-MoE architecture. The strategy aims to refine the representations of critical tokens by performing a secondary, partial forward pass. Our objective is to quantify the percentage increase in Floating-Point Operations (FLOPs) for a scenario with a sequence length of 16,384 tokens ($S = 16,384$), where 20% of tokens ($r = 0.2$) are selected for this recalculation. The DND mechanism is selectively applied to the 40 intermediate layers of the 48-layer model, excluding the initial four and final four layers.

The FLOPs for a single decoder layer are dominated by the self-attention mechanism and the Mixture-of-Experts (MoE) feed-forward network. The computational complexity of self-attention is quadratic with respect to sequence length ($\text{FLOPs}_{\text{attn}} \approx 4N_h d_h S^2$), while the MoE MLP complexity is linear ($\text{FLOPs}_{\text{moe}} \approx 6SkHI_{\text{moe}}$). The additional computation from the DND policy's nested pass, processing a fraction $r$ of the tokens, is therefore $\text{FLOPs}_{\text{added}} \approx r^2 \cdot \text{FLOPs}_{\text{attn}}(S) + r \cdot \text{FLOPs}_{\text{moe}}(S)$. The analysis is based on the parameters detailed in Tab. 5.

Table 5: Key Model and Strategy Parameters

| Parameter | Symbol | Value |
|---|---|---|
| Sequence Length (original) | $S$ | 16,384 |
| Hidden Size | $H$ | 2,048 |
| Number of Attention Heads | $N_h$ | 32 |
| Head Dimension | $d_h$ | 128 |
| Total Decoder Layers | $L_{\text{total}}$ | 48 |
| Layers with DND enabled | $L_{\text{dnd}}$ | 40 |
| MoE Intermediate Size | $I_{\text{moe}}$ | 768 |
| Activated Experts per Token | $k$ | 8 |
| Recalculation Ratio | $r$ | 0.20 |

For a standard layer operating on a 16k sequence, the self-attention component requires approximately $4.40 \times 10^{15}$ FLOPs, and the MoE MLP requires $1.24 \times 10^{15}$ FLOPs, totaling $5.64 \times 10^{15}$ FLOPs per layer. The additional computation from the nested pass is calculated as $(0.2)^2 \cdot (4.40 \times 10^{15}) + (0.2) \cdot (1.24 \times 10^{15}) \approx 0.424 \times 10^{15}$ FLOPs. This constitutes a per-layer overhead of 7.52% for the DND-enabled layers. When scaled across the entire model, the total computational overhead is proportional to the fraction of layers implementing the strategy.

$$\text{Overhead}_{\text{total}} = \left( \frac{\text{FLOPs}_{\text{added}}}{\text{FLOPs}_{\text{layer}}} \right) \times \frac{L_{\text{dnd}}}{L_{\text{total}}} = 7.52\% \times \frac{40}{48} \approx \mathbf{6.27\%}$$

In conclusion, the DND strategy with a 20% token recalculation ratio results in a modest total computational overhead of approximately **6.27%**. This increase in FLOPs should be weighed against the potential gains in model performance that the targeted recalculation may provide.

# B MORE DETAILS

**Hyper-parameters.** For the value of $\mathbf{L}_s$ and $\mathbf{L}_e$, we determined through ablation experiments that $\mathbf{L}_s = 4$ and $\mathbf{L}_e = 43$. In terms of the proportion of tokens to review, we balance performance and efficiency through ablation experiments on the Qwen1.7B model, ultimately setting the $k_{\text{target}}$ to 20%. For the initialization of the parameter $\beta$, to prevent overly gentle learning of the DND logic in the early training phase, we initialize it to 0.1. Regarding the hyperparameters defined in the training strategy loss function, due to the approximate 60x magnitude difference between the two loss extremes, we set $\lambda_{\text{sd}}$ to 3e-4 and $\lambda_{\text{dp}}$ to 0.02. The buffer size $N_b$ and the adjustment factor $\alpha$ are set to 5 and 5e-3 to balance stability and real-time performance, and $\gamma$ is set to 0.2. To ensure stability during initial training, we initialize token routers with all zeros and set the threshold to 0.5.

Table 6: **Differences between our DND approach and MOR.** Both methods aim to improve performance via increased computational depth, but differ significantly in implementation and control.

| Aspect | MOR | Ours (DND) |
|---|---|---|
| **Validation & Scaling** | | |
| **Application Stage** | Pre-training (Train from scratch). | Post-training (SFT). |
| **Model Scale** | Limited to models with around 1B parameters. | Applied from 1B dense to 30B MoE models. |
| **Purpose** | Achieve comparable performance with fewer parameters by using dynamically cycled depth under the same FLOPs budget. | Achieve better performance with almost unchanged parameter count by adding only a small amount of extra depth computation. |
| **Compatibility** | Requires from-scratch pre-training and substantial architectural changes when applying. | Strong compatibility; Can be directly applied to existing models post-training. |
| **Methodology** | | |
| **Architecture** | Dynamic depth output is treated as the final state. | Employs a normalized fusion strategy to combine dynamic and original outputs. |
| **Token Selection** | Relies on z-loss, an indirect load-balancing. | Introduces a precise ratio control method via discriminative routing output and dynamic thresholds. |
| **Control Precision** | Lacks precise control over the ratio of different depths. | Flexibly regulates the token selection proportion to any specified target. |

**Training Details.** The post-training stage uses the AdamW optimizer (Loshchilov & Hutter, 2017), with $\beta_1 = 0.9$, $\beta_2 = 0.95$, weight decay of 0.1, and gradient clipping at 1.0. During the post-training stage, we employ a cosine learning rate scheduler with a learning rate of $5 \times 10^{-6}$ that gradually decays to a minimum of $1 \times 10^{-6}$. All experiments are run on H800 GPUs. For Qwen3-1.7B, training is conducted for two epochs across 128 GPUs, taking one day. For Qwen3-30B-A3B, training is conducted for four epochs across 256 GPUs, taking approximately three days. For Llama-3.2-1B and Gemma3-1B, training is conducted for two epochs across 64 GPUs, taking approximately 1-2 days.

**Evaluation Metrics.** The evaluation includes several specialized testing protocols:

**AIME Evaluation:** On AIME24, we run inference 16 times per question for each model and report the average accuracy.
**IFEval Scoring:** The final score for IFEval is the average of the strict accuracies at both the prompt and instruction levels.

## C  DETAILED COMPARISONS WITH MOR

As shown in the Tab. 6, our DND differs significantly from MOR in terms of validated model scale, compatible training paradigms, model architecture, and token selection control strategies.

## D  MORE TOKEN SELECTION ANALYSIS

Beyond the entropy analysis of token logits in Sec. 4.5, we further examined another selection pattern of DND across layers. Specifically, we measured the variation in each token's hidden state across transformer layers. As shown in Fig. 8, we found that DND tends to select tokens whose representations change more strongly after passing through a certain transformer layer. This suggests that the model may be less confident about such tokens and therefore chooses to revisit them to obtain a more reliable representation.

## E  MORE VISUALIZATIONS

In the main text, we present visualizations of token selection by DND on general language tasks. In this section, we further provide visualizations for mathematics and coding tasks in Fig. 9 and Fig. 10. As shown, DND selectively allocates additional computation to key intermediate results and calculation steps in mathematics, while focusing on critical variables and essential logical comments in code. These observations align with our motivation and further validate the effectiveness of DND.

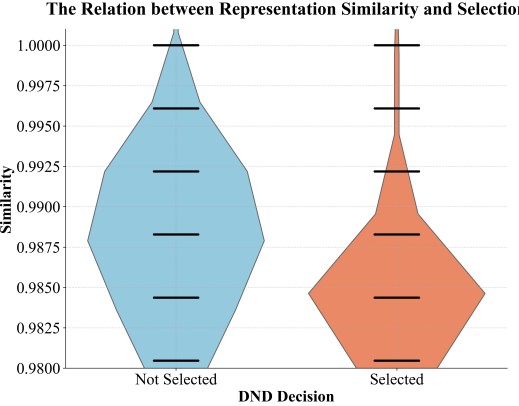

Figure 8: **Analysis of the Relationship between DND's Selection Preference and the Magnitude of Hidden State Changes across Transformer Layers.** DND exhibits a stronger preference for selecting tokens whose representations undergo larger changes after passing through a given layer.

**Question**: "Dave **bought** a large pack of french fries and ate **fourteen** before a hungry seagull stole the pack out of his hand. When the seagull landed, he gobbled down **half the amount** of french fries that Dave ate. Then three pigeons bullied him away from the food, and each pigeon **ate three fries**. Later, a raccoon stole **two thirds of** the **remaining** fries. Ants carried off **a final french fry**, leaving five behind. **How many** french fries were in the pack **when Dave bought** it?
**Let's think step by step.**
**Answer:"**

**Prediction: "Let's work through this step by step.**

**1. Let the total number of french fries in the pack be x.**
**2. Dave ate 14 fries. So, remaining after Dave: x - 14**
**3. The seagull ate half the amount Dave ate: Half of 14 = 7 fries So, seagull ate 7 fries. Remaining after seagull: (x - 14) - 7 = x - 21**
**4. Three pigeons each ate 3 fries: 3 pigeons × 3 fries = 9 fries eaten by pigeons Remaining after pigeons: (x - 21) - 9 = x - 30**
**5. A raccoon stole two-thirds of the remaining fries:**
**Raccoon took (2/3)(x - 30) So, two-thirds gone, one-third remains: Remaining after raccoon: (1/3)(x - 30)**
**6. Ants carried off 1 fry, leaving 5 behind: So, before ants took 1, there were 5 + 1 = 6 fries Therefore: (1/3)(x - 30) = 6**
**Now solve for x: Multiply both sides by 3: x - 30 = 18  x = 18 + 30  x = 48**

**So, the pack originally had 48 french fries.**
**Let's verify:- Start: 48 - Dave eats 14 → 48 - 14 = 34 - Seagull eats 7 → 34 - 7 = 27 - Pigeons eat 9 → 27 - 9 = 18- Raccoon steals 2/3 of 18 =**
**12 → 18 - 12 = 6 - Ants take 1 → 6 - 1 = 5 left All matches.**
**The answer is 48"**

Figure 9: **Visualization Results of A Mathematical Example.** Highlighted tokens are frequently selected, with orange indicating higher selection frequency in shallow layers and darker red representing higher selection in deeper layers.

## F  FAILURE CASE ANALYSIS

As shown in Fig. 11, we present a representative example in which Qwen3-30B-A3B answers correctly, but Qwen3-30B-A3B+DND fails. In this example, the task is to match foods to their corresponding meals based on the user's description. Under the DND "reviewing" critical tokens mode, the temporal cue **"Earlier"** in the prompt is excessively reviewed, which leads the model to overemphasize earlier times of day (roughly morning and noon). As a result, it incorrectly associates coffee with breakfast and pizza with lunch, and answers "breakfast and lunch" instead of the correct, unknown category "snack."

## G  THE USE OF LARGE LANGUAGE MODELS (LLMs)

Large language models (LLMs) were used as a writing aid in the preparation of this manuscript. Their use was strictly limited to improving grammar, phrasing, and formatting. All scientific content, including research ideas, methodology, data analysis, and conclusions, was conceived and developed exclusively by the authors. The authors have reviewed all AI-generated suggestions and take full responsibility for the final manuscript's accuracy and integrity.

```
Question:"You are given a sequence of N positive integers A=(A_1,A_2,\\dots,A_N).
Find the number of pairs of integers (l,r) satisfying 1\\leq l\\leq r\\leq N such that the subsequence
(A_l,A_{l+1},\\dots,A_r) forms an arithmetic progression.
A sequence (x_1,x_2,\\dots,x_{|x|}) is an arithmetic progression if and only if there exists a d such
that x_{i+1}-x_i=d\\ (1\\leq i < |x|).
In particular, a sequence of length 1 is always an arithmetic progression.
Enclose your code within delimiters as follows.
```python
# YOUR CODE HERE
```
### Answer: (use the provided format with backticks)"

Prediction: "```python
# Read input
N = int(input())
A = list(map(int, input().split()))

# Count arithmetic progressions
count = 0

# Every single element is an arithmetic progression
count += N

# Check all pairs of elements (length 2)
for i in range(N - 1):
    if A[i] == A[i + 1]:
        count += 1

# Check all subsequences of length >= 3
for l in range(N):
    for r in range(l + 2, N):  # length >= 3
        # Check if A[l:r+1] is arithmetic
        d = A[l + 1] - A[l]
        is_arithmetic = True
        for i in range(l + 1, r):
            if A[i + 1] - A[i] != d:
                is_arithmetic = False
                break
        if is_arithmetic:
            count += 1

print(count)
```"
```

Figure 10: **Visualization Results of A Code Generation Example.** Highlighted tokens are frequently selected, with orange indicating higher selection frequency in shallow layers and darker red representing higher selection in deeper layers.

```
{
    "gold": "[{'log_food': {'food_name': ['frozen mango'], 'meal_type': [ 'snack']}}, {'log_food': {'food_name': ['chai tea'],  'meal_type': ['snack']}}, {'log_food': {'food_name': ['pepperoni pizza'], 'meal_type': ['snack']}}, {'log_food': {'food_name': ['coffee'], 'meal_type': ['snack']}}]",

    "prediction": "[log_food(food_name="frozen mango", meal_type="snack"), log_food(food_name="chai tea", meal_type="snack"), log_food(food_name="pepperoni pizza", meal_type="lunch"), log_food(food_name="coffee", meal_type="breakfast")]",

    "origin_prompt": [
        {
            "content": ""You are given a list of foods someone ate today: apple (1 piece) and coffee (1 cup). For each food, decide which meal type it belongs to: breakfast, lunch, dinner, or snack. Answer with a Python list of strings, one for each food, in the same order.",
            "role": "system"
        },
        {
            "content": "I had 8 pieces of frozen mango and a chai tea.\n\n Earlier I had two slices of pepperoni pizza and a coffee",
            "role": "user"
        }
    ]
}
```

Figure 11: **Failure Case of DND.** DND may make errors by over-interpreting certain latent or weak pieces of evidence tokens.

## H LIMITATIONS AND FUTURE WORKS

- Our experiments primarily focus on validating the effectiveness of the DND strategy during the post-training stage of LLM. Its impact in pre-training or continual pre-training settings remains to be further explored.

- While we have applied DND in large auto-regressive models, exploring its applicability to other types of LLMs, such as diffusion-based LLMs, may be a promising direction for future research.

- Our subjective analyses reveal layer-wise preferences for different token types and show that the final token selection ratios vary across layers. Such observations could offer valuable insights for the design of future models.

