# OpenReview forum: "DND: Boosting Large Language Models with Dynamic Nested Depth"
_ICLR.cc/2026/Conference — ICLR 2026 Poster_

### Official Review · Reviewer_i7d3 · 2025-10-31

**Soundness:** 4
**Presentation:** 3
**Contribution:** 2
**Rating:** 4
**Confidence:** 3

**Summary:**

The paper presents a new heuristic method using token-conditional dynamic depth, applied post-training for improving accuracy at a low additional training and inference cost. The paper is aimed at LLM system engineers and inference optimization researchers, people who modify or extend pretrained transformer architectures to trade off accuracy and compute efficiency.

- Introduces Dynamic Nested Depth (DND): a router selects “hard” tokens, which are reprocessed once through the same transformer block; the outputs are fused with the original states via a normalized gate.
- Adds a router-control loss (to keep scores separable yet non-saturated) and a threshold-control loop (buffer proportional control + EMA) to stabilize the selected-token ratio.
- Applies DND to mid-layers of Qwen3-1.7B (dense) and Qwen3-30B-A3B (MoE) models; reports modest average accuracy gains with ~6 % FLOPs overhead and negligible parameter growth (< 0.1 M).

**Strengths:**

- Simple, reproducible architecture with minimal parameter overhead.
- Clear ablation analysis separating router and threshold controls.
- Good methodological transparency (layer range, selection ratios, FLOPs estimate).

**Weaknesses:**

- Small performance improvements, both in an absolute sense and relative to known model variants. Even fine tuning or just random variation might yield performance improvements similar to those shown in the paper.
- Weak experimental support: only within-model deltas, no compute-matched baselines, and no wall-clock profiling.
- Incremental and heuristic: conceptually similar to existing adaptive-depth ideas (MoD, MoR, ITT, early-exit) and lacks deeper theoretical insight.

**Questions:**

- Clarify whether base model weights were frozen or lightly fine-tuned during DND training.
- Include compute-matched comparisons with other adaptive-compute methods.
- Discuss failure modes—cases where DND reduces accuracy or increases instability.

---

> ### Author Response · Authors · 2025-11-19
> **Rebuttal (Part 1/4)**
>
> We appreciate your positive comments on our method and ablation studies,
> and we also appreciate your suggestions to evaluate more model architectures, report wall-clock inference speed,
> and compare with related approaches; we address each of these points in detail below.
>
>
> ### **W1. Accuracy Improvement**
>
> We appreciate your question regarding the magnitude of the performance improvements;
> in what follows, we explain in detail why these gains are substantial.
>
> **1. Robustness of gains:** First, to demonstrate that our method can provide substantial gains on small-scale LLMs,
> we additionally conducted experiments on Llama-3.2-1B-Base and Gemma3-1B-pt. We have also incorporated the supplemental experiments into the revised manuscript.
>
> As shown in the table below, compared with vanilla SFT, **applying DND to Llama-3.2-1B and Gemma3-1B leads to clear performance gains,
> the average improvements are 2.61 and 2.50 points, respectively**.
>
> (_Details can be referred to Sec. 4.3 and Tab.1 on page 7 of the revised manuscript._)
>
>
> ||| **General Knowledge & Alignment** |||| | **Math & STEM** || **Coding & Agent** ||||
> |:--|:--:|:--:|:---:|:--:|:--:|:---:|:--:|:--:|:--:|:--:|:--:|:--:|
> || **Average** | **BBH**| **PIQA** | **C-Eval** | **MMLU** | **IFEval** |**GPQA**| **GSM8K** |**MBPP**| **Human** | **BFCL**  | **MultiPLE** |
> | **Llama3.2-1B** |45.37|25.73 |   65.48   |   47.82    |   53.28   |   52.45    |      10.73      |   63.23   |       49.54        |   50.42   |   44.89   |    35.47     |
> | **+ DND**       |  **47.98**  |             **29.43**             | **66.51** | **49.21**  | **55.63** | **55.68**  |    **14.59**    | **66.57** |     **52.91**      | **52.16** | **47.52** |  **37.56**   |
> | *Δ (+-)*| ***+2.61*** |              *+3.70*              |  *+1.03*  |  *+1.39*   |  *+2.35*  |  *+3.23*   |     *+3.86*     |  *+3.34*  |      *+3.37*       |  *+1.74*  |  *+2.63*  |   *+2.09*    |
> | **Gemma3-1B**   |    47.08    |25.93  |   70.27   |   50.14    |   55.98   |   54.52    |      16.49      |   65.53   |       49.29        |   52.62   |   40.69   |    36.43     |
> | **+ DND** |  **49.58**  | **30.62**  | **71.33** | **51.00**  | **58.04** | **56.91**  |    **21.79**    | **68.68** |     **52.96**      | **53.76** | **42.88** |  **37.41**   |
> | *Δ (+-)* | ***+2.50*** |  *+4.69* |  *+1.06*  |  *+0.86*   |  *+2.06*  |  *+2.39*   |     *+5.30*     |  *+3.15*  |      *+3.67*       |  *+1.14*  |  *+2.19*  |   *+0.98*    |
>
> Therefore, **across more than ten mainstream benchmarks covering diverse tasks—including general, code, math, reasoning, agent, Chinese,
> and alignment—our DND yields average accuracy improvements of 2.50, 2.61, and 1.88 points on Gemma3-1B, Llama-3.2-1B, and Qwen3-1.7B, respectively.**
> These results show that the gains from our approach are substantial and are not restricted to specific LLMs or task categories.
>
> **2. Scalability of gains:** In scaling experiments on the 30B-parameter Qwen3-30B-A3B model, applying our DND strategy during SFT yields an average improvement of close to 1 point across nearly 20 benchmarks.
> In particular, on challenging code and reasoning benchmarks such as **GPQA, HumanEval+, MultiPLE, LiveCodev5, LiveCodev6, and BFCL v3, we observe gains of 0.91, 0.99, 1.12, 1.24, 1.42, and 2.05**, respectively.
> Moreover, we do not observe performance degradation on any benchmark, indicating that our method scales effectively to current large-parameter MoE models.
>
> **3. Low cost and ease of reproduction:** The gains from our method come at relatively low cost,
> since it does not require any changes during pretraining or continued pretraining.
> Instead, the **DND mechanism can be directly applied in the SFT stage** of any fully pretrained base model to obtain stable performance improvements.
> In addition, DND is compatible with mainstream LLMs without modifying other architectures, which lowers the barrier to adoption and replication.
>
> **4. Variation evaluation:**  To directly address your concern that simple random perturbations might yield similar improvements,
> we conducted a training-free ablation on Qwen3-1.7B.
> In this experiment, we selected for review the top 20% of the prefilled tokens whose hidden states changed the most after passing through each transformer layer,
> based on the intuition and our observations that larger changes in the initial forward pass reflect higher learning difficulty.
> However, we only observed fluctuations in performance across datasets without consistent gains.
> **This indicates that a simple variation baseline cannot reproduce the performance improvements achieved by our DND method.**

---

> ### Author Response · Authors · 2025-11-19
> **Rebuttal (Part 2/4)**
>
> ### **W2.1. Wall-Clock Analysis**
>
> We appreciate your suggestion to report actual throughput in addition to FLOPs.
> Following your advice, we adopted the throughput reporting standards from the authoritative LLM inference acceleration framework, SGlang.
> During inference, we benchmarked Qwen3-30B-A3B and Qwen3-30B-A3B+DND on a single H100 GPU with batch size 1 and BF16 precision, and reported throughput as **tokens per second (TPS)**, defined as the total number of prefill and decode tokens divided by the total prefill plus decode time.
> Besides, input and output lengths of LLMs were set according to SGlang report standards.
>
> We evaluated throughput under four scenarios (input lengths of 1K and 6K, output lengths of 2K and 6K), representing short and long input/output conditions.
> As shown in the table below, **our model achieves 91.6–93.1% of the baseline throughput across all cases**,
> indicating that the addition of DND does not significantly reduce inference speed.
>
> (_Details can be referred to Sec. 4.3 and Tab.3 on page 8 of the revised manuscript._)
>
> | Input Length | Decode Length | Speed (tokens/s)  | Speed (tokens/s) | Relative Speed (%) |
> |:----:|:-----:|:--:|:-:|:-:|
> | || **Qwen3-30B-A3B** |**+ DND** ||
> | 1024|2048|148.68|136.19 | 91.6 |
> | 6144|2048|208.60  |193.51 | 92.8 |
> | 1024|6144|76.16 | 70.52  | 92.6 |
> | 6144|6144|100.89 |93.93 | 93.1  |
>
> ### **Q1. Details of SFT**
>
> We appreciate your valuable suggestion. In the revised manuscript, we have added more training details for SFT.
> In all experiments, we **used standard SFT, where all model parameters were trainable**, including the shallowest and deepest transformer layers that do not apply the DND strategy.
>
> (_Details can be referred to Sec. 4.2 on page 7 of the revised manuscript._)
>
> ### **W3.1. Novelty Clarification**
>
> We would like to clarify that, to the best of our knowledge,
> **DND is the first method that is suitable for post-training on autoregressive LLMs, which improves performance by dynamically increasing computation depth.**
> While we acknowledge and respect the contributions of MoD, MoR, ITT, and early-exit, and agree that they all dynamically adjust the computation depth of transformer layers, they are fundamentally different from DND.
>
> MoD and early-exit reduce computation depth to speed up inference.
> ITT also increases computation depth dynamically, but its top-p routing design can cause information leakage,
> which makes it unsuitable for autoregressive LLMs.
> The most similar work, MoR, trains models from scratch during pretraining with fewer initialized layers,
> and then uses dynamic increases in computation depths to obtain similar performance with fewer parameters under the same FLOPs budget.
> However, this strategy cannot be directly applied to existing open-source LLMs,
> because removing any pretrained layer would severely damage the pretrained knowledge.
>
> (_Details can be referred to Appendix Sec. C and Tab.6 of the revised manuscript._)
>
> In contrast, DND is designed as a post-training method that can be directly applied to current open-source LLMs.
> We have verified that **DND brings stable gains on 1B–2B dense LLMs** such as Qwen3, Llama-3.2, and Gemma3,
> and **we have further scaled it to a 30B-parameter MoE model, which is, to our knowledge, the first successful attempt of all dynamic depth methods at this scale**.
>
> DND, together with these similar methods, can be viewed as **studies of the vertical architecture of transformers**.
> Compared with past work that mainly focuses on horizontal architecture, such as various MoE variants,
> vertical-depth methods that increase computation along the layer axis are emerging concurrently with our work,
> suggesting that the community is beginning to pay more attention to vertical design.
> We believe that DND provides both confidence and useful insights for further research along this direction.

---

> ### Author Response · Authors · 2025-11-19
> **Rebuttal (Part 3/4)**
>
> ### **W2.2. & Q2.Computation-matched Methods Comparison**
>
> We appreciate your valuable suggestion to compare our method with other approaches that also increase computation.
> Since the only existing work on dynamically increasing computation in LLMs is ITT,
> we reproduced ITT under the same training setup using Qwen3-1.7B for comparison.
>
> As shown in the table below, when SFT is conducted with ITT under the same computation budget,
> the performance improvement is less pronounced.
> We attribute these limited gains to the use of top-p–based token selection in the autoregressive setting,
> which introduces a mismatch between training and inference and may also lead to potential information leakage.
>
> (_Details can be referred to Sec. 4.3 and Tab.1 on page 7 of the revised manuscript._)
>
> ||**Average**|**BBH**|**PIQA**|**C-Eval**|**MMLU**|**IFEval**|**GPQA**|**GSM8K**|**MBPP** |**Human+**|**BFCL**| **MultiPLE** |
> |:--:|:--:|:--:|:--:|:--:|:--:|:--:|:--:|:--:|:--:|:--:|:--:|:---:|
> |Qwen3-1.7B | 59.53 | 40.82 | 75.38 | 60.00|64.11 |65.47| 28.54|79.38|68.38 |61.59|58.73|52.45|
> |+ ITT|59.58 | 41.23 | 75.99 | 59.51 | 64.92 | 65.08 |  27.85| 80.29| 68.67 | 60.78 | 58.62 |52.45|
> | **+ DND** |**61.41** | **45.84**| **76.25** | **60.38**| **64.45**| **66.87** |**34.34** | **80.15**| **71.90** |**62.71** | **59.80** |  **52.80**|
> |*$\Delta$ (+-)*|*+1.88*|*+5.02* |*+0.87*|*+0.38* | *+0.34*|*+1.40* |*+5.80*|  *+0.77*|  *+3.52*  |*+1.12*|*+1.07*| *+0.35*  |
>
> ### **W3.2. More Insight**
>
> We appreciate your comment regarding the need for deeper insights.
> Following reviewer pkVU’s suggestion, we conducted a more detailed analysis of the characteristics of the selected tokens,
> as well as a deeper analysis into why the DND mechanism is able to yield consistent performance improvements.
>
> **1. Why the selected tokens are critical?**
>
> Based on your suggestion, we analyzed the relationship between the number of times each token is selected by the DND router for nested depth review and the entropy of the token’s vanilla logits.
> Specifically, we conducted this analysis by analyzing Qwen3-30B-A3B+DND on the MMLU dataset.
> Our results **show a clear positive correlation between selection frequency and the vanilla logit entropy**,
> with a Pearson coefficient of approximately 0.34 (where 1 indicates perfect correlation and -1 indicates perfect negative correlation).
> In this context, the selection frequency indicates how many times a token is routed for review by the DND mechanism across multiple transformer layers.
>
> Prior research demonstrates that tokens with high logit entropy correspond to tokens for which the model is uncertain or hesitant between multiple possible outcomes, _i.e._,
> tokens that are inherently difficult for the LLM to learn.
> Therefore, these results strongly support the motivation behind our approach and demonstrate that our router does not select tokens randomly;
> rather, **it systematically prefers harder-to-learn tokens**.
>
> (_Details can be referred to Sec. 4.5.1 and Fig.4a on page 9 of the revised manuscript._)
>
> **2. Why are the representations of these tokens better learned under the DND mechanism?**
>
> Furthermore, to demonstrate that our DND strategy effectively enables the model to learn better representations for these critical tokens,
> we analyzed the variation in a token’s logit entropy between the vanilla forward pass and the DND forward pass.
> We found **a clear positive correlation between the reduction in a token’s logit entropy and the number of review times**, with a Pearson coefficient of -0.58.
>
> Specifically, when a token was selected only a few times (fewer than three), the change in its logit entropy was roughly symmetric in both directions.
> However, when the token was selected more than ten times, the logit entropy showed a pronounced decreasing trend,
> continuing to decrease as the number of reviews increased.
> This experiment strongly supports that our **DND strategy effectively increases the LLM’s confidence in these critical tokens, enabling more certain and less hesitant representations**.
>
> (_Details can be referred to Sec. 4.5.2 and Fig.4b on page 9 of the revised manuscript._)
>
> **3. Other observations.**
>
> Additionally, we analyzed another selection tendency of DND across layers.
> We analyzed the degree of variation in hidden states for each token at different transformer layers and found that
> **DND tends to select tokens whose representations change more strongly across layers**.
> This may be due to the model having lower confidence in those tokens whose representations change greatly across layers, so it needs to review these tokens to obtain a more reliable representation.
>
> (_Details can be referred to Appendix Sec. D and Fig.8 of the revised manuscript._)
>
> **To the best of our knowledge, DND is the first work on dynamic-depth LLMs that provides an explicit quantitative
> analysis of the properties of selected tokens and empirically shows why dynamic depth improves LLM performance.**

---

> ### Author Response · Authors · 2025-11-19
> **Rebuttal (Part 4/4)**
>
> ### **Q3. Failure Case Analysis**
>
> We appreciate your question regarding failure modes. To address your concern, we selected a representative case from the
> BFCL-v3 dataset for detailed analysis, and we have incorporated this analysis into the revised manuscript.
>
> As shown in the case below, we present a representative example in which Qwen3-30B-A3B answers correctly,
> but Qwen3-30B-A3B+DND fails. In this example, the task is to match foods to their corresponding meals based on the user's description.
> Under the DND “reviewing” critical tokens mode, the temporal cue “**Earlier**” in the prompt is excessively reviewed,
> which leads the model to overemphasise earlier times of day (roughly morning and noon).
> As a result, it incorrectly associates coffee with breakfast and pizza with lunch,
> and answers “breakfast and lunch” instead of the correct, unknown category “snack.”
>
> (_Details can be referred to Appendix Sec. F and Fig.11 of the revised manuscript._)
>
> ```json
> {
>   "gold": " 'food_name': ['frozen mango'], 'meal_type': ['snack'], 'food_name': ['chai tea'],  'meal_type': ['snack'], 'food_name': ['pepperoni pizza'], 'meal_type': ['snack'], 'food_name': ['coffee'], 'meal_type': ['snack'] ",
>
>   "prediction": " 'food_name': ['frozen mango'], 'meal_type': ['snack'], 'food_name': ['chai tea'],  'meal_type': ['snack'], 'food_name': ['pepperoni pizza'], 'meal_type': ['lunch'], 'food_name': ['coffee'], 'meal_type': ['breakfast'] ",
>
>   "origin_prompt": [
>     {
>       "content": "You are given a list of foods someone ate today: apple (1 piece) and coffee (1 cup). For each food, decide which meal type it belongs to: breakfast, lunch, dinner, or snack. Answer with a Python list of strings, one for each food, in the same order.",
>       "role": "system"
>     },
>     {
>       "content": "I had 8 pieces of frozen mango and a chai tea. Earlier I had two slices of pepperoni pizza and a coffee",
>       "role": "user"
>     }
>   ]
> }
> ```

---

> ### Author Response · Authors · 2025-11-26
>
> Dear Reviewer i7d3,
>
> We hope that our follow-up response and the supplementary experiments help address your questions. Your review is greatly appreciated, and we welcome any additional feedback you may have. If any part of our response—or any other point—remains unclear, please let us know, and we would be happy to elaborate further.
>
> Thank you for your valuable time!
>
> Sincerely,
>
> Authors of Paper 8392

---

> ### Author Response · Authors · 2025-11-27
>
> Dear Reviewer i7d3,
>
> We are truly grateful for your thorough review and the time you have invested in evaluating our work. We sincerely hope that our follow-up response, along with the supplementary experiments, has effectively addressed your questions. Your feedback is incredibly valuable to us, and we would deeply appreciate any additional thoughts or suggestions you may have.
>
> If any part of our response—or any other point—remains unclear, please don’t hesitate to let us know. We would be more than happy to provide further clarification or elaboration as needed.
>
> Thank you once again for your precious time and thoughtful consideration.
>
> Warm regards,
>
> Authors of Paper 8392

---

> ### Author Response · Authors · 2025-11-28
>
> Dear Reviewer i7d3,
>
> We appreciate the time you dedicated to reviewing our work. We hope that our response and additional experiments have clarified your concerns. Please let us know if any points remain unclear; we would be happy to elaborate further.
>
> Thank you once again for your time and consideration.
>
> Best regards,
>
> Authors of Paper 8392

---

### Official Review · Reviewer_o89C · 2025-10-31

**Soundness:** 2
**Presentation:** 2
**Contribution:** 2
**Rating:** 4
**Confidence:** 4

**Summary:**

The paper introduces a method called Dynamic Nested Depth (DND) which adaptively identifies “difficult” tokens for layer in an LLM and allocates extra computation to those tokens by re‑processing them through the same layer in a nested way, while easier tokens receive standard processing. This token‑level routing is managed by a lightweight router that assigns each token a probability of being selected using hidden state of each token in the seqeunce; if the score exceeds a threshold, the token undergoes a “nested” pass. Once processed, the outputs of the first pass and the subsequent passes are merged together before moving to the next layer.

**Strengths:**

- The problem formulation is compelling, as allocating additional computation to difficult tokens could improve model accuracy at the cost of additional compute.
 - Experiments are adequate and includes evaluations across diverse benchmarks (knowledge, reasoning, coding).
 - Analysis is insightful, with clear plots and ablations that illustrate how token-level nested depth impacts performance.

**Weaknesses:**

- The reported accuracy improvement over regular SFT is only 0.87 on average in table 1, which appears minimal. The proposed method involves processing the input multiple times for some tokens in the given layers. Such process would increase the compute overhead. But this table does not include the computational overhead introduced to bring the 0.87 avg accuracy improvement. This makes the experimental results incomplete.
 -  Experiments are conducted on only a single LLM (Qwen), raising concerns about the generalizability of the method to other architectures.

-  Based on the method description and experiments, it is unclear whether DND fine-tunes all components of the transformer blocks in the selected layers (L_s to L_e) or if it fine-tunes all model parameters like standard SFT.

- The router-level losses and their underlying motivation are not clearly explained. Moreover, their overall contribution appears minimal, as reflected by the similar values in Table 2 (60.54, 60.58, 60.64) where these losses are not included.

 - Overall, the approach seems largely heuristic and does not demonstrate a substantial improvement in accuracy.

**Questions:**

Based on the method description and experiments, it is unclear whether DND fine-tunes all components of the transformer blocks in the selected layers (L_s to L_e) or if it fine-tunes all model parameters like standard SFT.

---

> ### Author Response · Authors · 2025-11-19
> **Rebuttal (Part 1/2)**
>
> We sincerely thank you for your recognition of the soundness of our method and the insightfulness of our analysis.
> We also appreciate your suggestions to include inference cost measurements and to evaluate DND on a more diverse set of LLMs.
> Additionally, we will further clarify the obvious performance gains brought by DND,
> the rationale and effects of the proposed router loss, and the details of SFT.
>
> ### **W1. Compute Overhead**
>
> We appreciate your suggestion to include a discussion on compute overhead.
> We have also incorporated this discussion into the revised manuscript.
>
> Firstly, we would like to clarify that in Appendix A, we have already detailed the additional FLOPs introduced by our method.
> For the Qwen3-30B-A3B+DND model, this amounts to approximately **6.27%** extra computation.
> Moreover, following your suggestion, we have also conducted experiments measuring the actual computational overhead.
>
> Specifically, we adopted the throughput reporting standards from the authoritative LLM inference acceleration framework, SGlang.
> During inference, we benchmarked Qwen3-30B-A3B and Qwen3-30B-A3B+DND on a single H100 GPU with batch size 1 and BF16 precision, and reported throughput as **tokens per second (TPS)**, defined as the total number of prefill and decode tokens divided by the total prefill plus decode time.
> Besides, input and output lengths of LLMs were set according to SGlang report standards.
>
> We evaluated throughput under four scenarios (input lengths of 1K and 6K, output lengths of 2K and 6K), representing short and long input/output conditions.
> As shown in the table below, **our model achieves 91.6–93.1% of the baseline throughput across all cases**,
> indicating that the addition of DND does not significantly reduce inference speed.
>
> (_Details can be referred to Sec. 4.3 and Tab.3 on page 8 of the revised manuscript._)
>
> | Input Length | Decode Length | Speed (tokens/s)  | Speed (tokens/s) | Relative Speed (%) |
> |:------------:|:-------------:|:-----------------:|:----------------:|:------------------:|
> |              |               | **Qwen3-30B-A3B** |    **+ DND**     |                    |
> |     1024     |     2048      |      148.68       |      136.19      |        91.6        |
> |     6144     |     2048      |      208.60       |      193.51      |        92.8        |
> |     1024     |     6144      |       76.16       |      70.52       |        92.6        |
> |     6144     |     6144      |      100.89       |      93.93       |        93.1        |
>
>
> ### **W2. Architecture Dependency**
>
> We appreciate your suggestion to evaluate DND on LLMs beyond our original architecture.
> We have also incorporated the supplemental experiments into the revised manuscript.
>
> Following this suggestion, we added experiments on two additional widely used pretrained models, **Llama-3.2-1B-Base** and **Gemma3-1B-pt**, as baselines.
> Each experiment was run on 64 H800 GPUs for 1–2 days.
> We then compared the performance of the vanilla models and their DND-enhanced counterparts across a diverse set of benchmarks.
>
> As shown in the table below, applying DND to Llama-3.2-1B and Gemma3-1B leads to clear performance gains.
> Across more than ten benchmarks—including general tasks, math, coding, agent,
> and alignment tasks—the average improvements are **2.61** and **2.50** points, respectively.
> **These results indicate that the benefits of DND are not restricted to the Qwen family of models.**
>
> (_Details can be referred to Sec. 4.3 and Tab.1 on page 7 of the revised manuscript._)
>
> | |  | **General Knowledge & Alignment** | | | | | **Math & STEM** | | **Coding & Agent** | | | |
> |:------|:----:|:--------:|:---------:|:------:|:---:|:-----:|:----:|:-----:|:----:|:----:|:----:|:--:|
> |  | **Average** |   **BBH**  | **PIQA**  | **C-Eval** | **MMLU**  | **IFEval** |    **GPQA**   | **GSM8K** |  **MBPP**   | **Human+** | **BFCL**  | **MultiPLE** |
> | **Llama3.2-1B** |    45.37    |   25.73   |   65.48   |   47.82    |   53.28   |   52.45    | 10.73 |   63.23   |  49.54 |   50.42   |   44.89   |    35.47     |
> | **+ DND**       |  **47.98**  |    **29.43**   | **66.51** | **49.21**  | **55.63** | **55.68**  |    **14.59**    | **66.57** |     **52.91**      | **52.16** | **47.52** |  **37.56**   |
> | *Δ (+-)*        | ***+2.61*** |  *+3.70*  |  *+1.03*  |  *+1.39*   |  *+2.35*  |  *+3.23*   |  *+3.86*  |  *+3.34*  |      *+3.37*       |  *+1.74*  |  *+2.63*  |   *+2.09*    |
> | **Gemma3-1B**   |    47.08    |25.93 |   70.27   |   50.14    |   55.98   |   54.52    |  16.49   |   65.53   |       49.29        |   52.62   |   40.69   |    36.43     |
> | **+ DND**       |  **49.58**  |   **30.62**     | **71.33** | **51.00**  | **58.04** | **56.91**  |    **21.79**    | **68.68** |     **52.96**      | **53.76** | **42.88** |  **37.41**   |
> | *Δ (+-)*        | ***+2.50*** | *+4.69*  |  *+1.06*  |  *+0.86*   |  *+2.06*  |  *+2.39*   |     *+5.30*     |  *+3.15*  |      *+3.67*       |  *+1.14*  |  *+2.19*  |   *+0.98*    |

---

> ### Author Response · Authors · 2025-11-19
> **Rebuttal (Part 2/2)**
>
> ### **W3. & Q1. Details of SFT**
>
> We appreciate your valuable suggestion. In the revised manuscript, we have added more training details for SFT.
>
> In all experiments, we used **standard SFT, where all model parameters were trainable**, including the shallowest and deepest transformer layers that do not apply the DND strategy.
>
> For Qwen3-30B-A3B+DND, we trained on 256 H800 GPUs for approximately 3–4 days with a per-GPU batch size of 4.
> For Qwen3-1.7B+DND, we used 128 H800 GPUs for about 1 day with a per-GPU batch size of 8.
> For Gemma3-1B and Llama-3.2-1B, we trained on 64 H800 GPUs for 1–2 days with a per-GPU batch size of 8.
>
> (_Details can be referred to Sec. 4.2 on page 7 of the revised manuscript._)
>
> ### **W4. The Motivations and Effectiveness of Router Control Loss**
>
> We appreciate your questions regarding the motivation and effectiveness of router control.
> We have clarified these points in the revised manuscript, and we provide a detailed explanation below.
>
> **Motivation:** Our goal is to identify critical tokens in the input sequence for review, while keeping the notion of “criticalness” flexible and adjustable.
> Because autoregressive LLMs cannot access future tokens during decoding, using a top-k/top-p selection scheme would lead to information leakage.
> **To avoid information leakage while maintaining flexibility**, we propose a proportion-based selector that consists of a linear router and a learnable threshold.
> We further introduce a **router-specific control loss to enhance the discriminability of routing decisions across tokens**,
> and we employ a **real-time proportion feedback mechanism to dynamically update the threshold, which helps maintain stable training**.
> The real-time feedback mechanism helps prevent large fluctuations in the selection proportion that could damage the pretrained knowledge in the early stages of training.
>
> **Effectiveness:** Because token selection is jointly determined by both the threshold and the router,
> relying on only one of them—threshold-only or router-only—results in incomplete control and limited performance gains,
> as shown in our ablation studies.
> In contrast, **the combined control strategy leverages the strengths of both components**,
> effectively regulating the selection proportion and yielding clear improvements.
> Across more than ten benchmarks, this approach achieves an average gain of nearly 1 point (from 1.05 or 1.15 to 1.88),
> which demonstrates its effectiveness.
>
> (_Clearer analysis of the effectiveness can be referred to Sec. 4.4 on page 8 of the revised manuscript._)
>
> ### **W5. Accuracy Improvement**
>
> We appreciate your question regarding the magnitude of the performance improvements;
> in what follows, we explain in detail why these gains are substantial.
>
> **1. Robustness of gains:** **Across more than ten mainstream benchmarks** covering diverse tasks—including general, code, math, reasoning, agent, Chinese,
> and alignment—our method **yields average accuracy improvements of 2.50, 2.61, and 1.88 points on Gemma3-1B, Llama-3.2-1B, and Qwen3-1.7B, respectively**.
> These results show that the improvements brought by our approach are **not limited to specific model families or task types**.
>
> **2. Scalability of gains:** In scaling experiments on the 30B-parameter Qwen3-30B-A3B model, applying our DND strategy during SFT yields an average improvement of close to 1 point across nearly 20 benchmarks.
> In particular, on challenging code and reasoning benchmarks such as **GPQA, HumanEval+, MultiPLE, LiveCodev5, LiveCodev6, and BFCL v3, we observe gains of 0.91, 0.99, 1.12, 1.24, 1.42, and 2.05, respectively**.
> We do not observe performance degradation on any benchmark, indicating that our method **scales effectively to current large-parameter MoE models**.
>
> **3. Low cost and ease of reproduction:** The gains from our method come at relatively low cost,
> since it does not require any changes during pretraining or continued pretraining.
> Instead, the DND mechanism can **be directly applied in the SFT stage of a fully pretrained base model** to obtain stable performance improvements.
> In addition, DND is compatible with mainstream LLMs **without modifying other architectures**, which lowers the barrier to adoption and replication.

---

> ### Author Response · Authors · 2025-11-26
>
> Dear Reviewer o89C,
>
> We hope our follow-up response and supplementary experiments help to address your questions. Your review is greatly valued, and we would appreciate any additional feedback you may have. Of course, please let us know if this or any other point remains unclear, as we would be happy to elaborate further.
>
> Thank you for your valuable time!
>
> Sincerely,
>
> Authors of Paper 8392

---

> ### Author Response · Authors · 2025-11-27
>
> Dear Reviewer o89C,
>
> We sincerely appreciate the time and effort you have put into reviewing our paper. We hope that our follow-up response and supplementary experiments have adequately addressed your concerns. Your feedback is of great importance to us, and we would be truly grateful for any further insights you might share.
>
> If any points remain unclear or if further clarification is needed, please don't hesitate to let us know. We are more than willing to elaborate and provide any additional information.
>
> Thank you once again for your invaluable time and thoughtful consideration.
>
> Warm regards,
>
> Authors of Paper 8392

---

> ### Author Response · Authors · 2025-11-28
>
> Dear Reviewer o89C,
>
> We sincerely appreciate your time and review. We hope our follow-up response and new experiments have resolved your questions. We are happy to provide further elaboration if any points remain unclear.
>
> Thank you once again for your time and consideration.
>
> Best regards,
>
> Authors of Paper 8392

---

### Official Review · Reviewer_pkVU · 2025-11-01

**Soundness:** 3
**Presentation:** 3
**Contribution:** 3
**Rating:** 6
**Confidence:** 3

**Summary:**

This paper proposes Dynamic Nested Depth (DND) — an efficient method to enhance pre-trained LLMs by selectively deepening computation. Instead of uniformly processing all tokens, DND identifies critical tokens that require more reasoning effort. Within each transformer layer, a lightweight router marks these important tokens after a standard forward pass. Only those tokens undergo an additional "nested" pass through the same layer, and their outputs are then fused with the initial results. DND can be applied in post-training (e.g., fine-tuning), making it compatible with existing models.

**Strengths:**

S1: While adaptive computation is an established research area, DND mechanism is a novel implementation. Instead of skipping layers or routing to different experts, the idea of re-processing a selected subset of tokens through the same transformer layer ("nested pass") is an elegant formulation, reminding me of recent looped Transformers. Furthermore, the decision to apply this technique during a post-training (SFT) phase is a highly original and pragmatic choice.

S2: Thorough Empirical Evaluation: The experiments are comprehensive and convincing. The method is validated on two different model scales and architectures (a dense 1.7B and a sparse MoE 30B model), and tested across 17 diverse benchmarks. The performance gains, especially in complex reasoning domains like coding and mathematics, are significant and consistently positive.

S3: The significance of this work is substantial, as it addresses one of the most pressing challenges in the development of LLMs: the trade-off between performance and computational cost.

**Weaknesses:**

W1: While this paper presents a promising direction,the paper does not quantitatively demonstrate why the selected tokens are "critical." The visualizations could be the result of cherry-picking or confirmation bias. It is unclear if the router has learned a meaningful selection strategy or is simply responding to surface-level statistical patterns. To strengthen this claim, the authors should perform a quantitative analysis correlating token selection with intrinsic properties of the tokens or the model's state.

For instance: Correlation with Model Uncertainty: Are the selected tokens ones for which the model has high prediction entropy or low softmax probability in the initial pass? This would suggest DND is focusing computation on "hard" decisions.

W2: The paper claims a "minimal parameter and computing increase," basing its efficiency analysis on a theoretical calculation of FLOPs.  FLOPs are a poor proxy for actual inference speed (latency and throughput) on GPUs. The DND architecture, with its Pack and Unpack operations, breaks the massive parallelism that GPUs are optimized for, potentially leading to significant latency increases that are not captured by FLOP counts. They should report: Wall-clock latency (ms per generated token) and throughput (tokens per second) on a standard benchmark GPU (e.g., A100 or H100).

**Questions:**

Q1: All experiments are conducted on a single family of models (Qwen3). While the results are positive, this makes it impossible to know if the DND approach is a general-purpose technique or if its effectiveness relies on specific, undocumented properties of the Qwen architecture.

---

> ### Author Response · Authors · 2025-11-19
> **Rebuttal (Part 1/2)**
>
> We sincerely thank you for your recognition of our work’s novelty and its strong performance improvements. And thank you for the valuable suggestions provided. We have carefully addressed each of the issues you raised and improved the work accordingly.
> We first designed experiments to demonstrate the motivation and effectiveness of DND’s critical-token selection mechanism, and we added a detailed comparison of throughput, along with additional experimental results on both the Llama and Gemma model families. The specifics are presented below.
>
> ### **W1. Validation of the Critical Token Selection**
>
> We thank you for your suggestion to include quantitative experiments validating the effectiveness of our DND token selection mechanism.
> Following your advice, we have added experiments to demonstrate **why the selected tokens are critical** and **why the representations of these tokens are better learned under the DND mechanism**.
> We have also incorporated these discussions into the revised manuscript.
>
>
> **1. Why the selected tokens are critical?**
>
> Based on your suggestion, we analyzed the relationship between the number of times each token is selected by the DND router for nested depth review and the entropy of the token’s vanilla logits.
> Specifically, we conducted this analysis by analyzing Qwen3-30B-A3B+DND on the MMLU dataset.
> Our results **show a clear positive correlation between selection frequency and the vanilla logit entropy**,
> with a Pearson coefficient of approximately 0.34 (where 1 indicates perfect correlation and -1 indicates perfect negative correlation).
> In this context, the selection frequency indicates how many times a token is routed for review by the DND mechanism across multiple transformer layers.
>
> Prior research demonstrates that tokens with high logit entropy correspond to tokens for which the model is uncertain or hesitant between multiple possible outcomes, _i.e._,
> tokens that are inherently difficult for the LLM to learn.
> Therefore, these results strongly support the motivation behind our approach and demonstrate that our router does not select tokens randomly;
> rather, **it systematically prefers harder-to-learn tokens**.
>
> (_Figures and Details can be referred to Sec. 4.5.1 and Fig.4a on page 9 of the revised manuscript._)
>
> **2. Why are the representations of these tokens better learned under the DND mechanism?**
>
> Furthermore, to demonstrate that our DND strategy effectively enables the model to learn better representations for these critical tokens,
> we analyzed the variation in a token’s logit entropy between the vanilla forward pass and the DND forward pass.
> We found **a clear positive correlation between the reduction in a token’s logit entropy and the number of review times**, with a Pearson coefficient of -0.58.
>
> Specifically, when a token was selected only a few times (fewer than three), the change in its logit entropy was roughly symmetric in both directions.
> However, when the token was selected more than ten times, the logit entropy showed a pronounced decreasing trend,
> continuing to decrease as the number of reviews increased.
> This experiment strongly supports that our **DND strategy effectively increases the LLM’s confidence in these critical tokens, enabling more certain and less hesitant representations**.
>
> (_Figures and Details can be referred to Sec. 4.5.2 and Fig.4b on page 9 of the revised manuscript._)
>
> **3. Other observations.**
>
> Additionally, we analyzed another selection tendency of DND across layers.
> We analyzed the degree of variation in hidden states for each token at different transformer layers and found that
> **DND tends to select tokens whose representations change more strongly across layers**.
> This may be due to the model having lower confidence in those tokens whose representations change greatly across layers,
> so it needs to review these tokens to obtain a more reliable representation.
>
> (_Figures and Details can be referred to Appendix Sec .D and Fig.8 of the revised manuscript._)

---

> ### Author Response · Authors · 2025-11-19
> **Rebuttal (Part 2/2)**
>
> ### **W2. Throughput Report**
>
> We appreciate your suggestion to report actual throughput in addition to FLOPs.
> We have also incorporated the throughput analysis into the revised manuscript.
>
> Following your advice, we adopted the throughput reporting standards from the authoritative LLM inference acceleration framework, SGlang.
> During inference, we benchmarked Qwen3-30B-A3B and Qwen3-30B-A3B+DND on a single H100 GPU with batch size 1 and BF16 precision,
> and reported throughput as **tokens per second (TPS)**, defined as the total number of prefill and decode tokens divided by the total prefill plus decode time.
> Besides, input and output lengths of LLMs were set according to SGlang report standards.
>
> We evaluated throughput under four scenarios (input lengths of 1K and 6K, output lengths of 2K and 6K), representing short and long input/output conditions.
> As shown in the table below, **our model achieves 91.6–93.1% of the baseline throughput across all cases**,
> indicating that the addition of DND does not significantly reduce inference speed.
>
> (_Details can be referred to Sec. 4.3 and Tab.3 on page 8 of the revised manuscript._)
>
> | Input Length | Decode Length | Speed (tokens/s)  | Speed (tokens/s) | Relative Speed (%) |
> |:------------:|:-------------:|:-----------------:|:----------------:|:------------------:|
> |              |               | **Qwen3-30B-A3B** |    **+ DND**     |                    |
> |     1024     |     2048      |      148.68       |      136.19      |        91.6        |
> |     6144     |     2048      |      208.60       |      193.51      |        92.8        |
> |     1024     |     6144      |       76.16       |      70.52       |        92.6        |
> |     6144     |     6144      |      100.89       |      93.93       |        93.1        |
>
>
> ### **Q1. Architecture Dependency**
>
> We appreciate your suggestion to evaluate DND on LLMs beyond our original architecture.
> We have also incorporated the supplemental experiments into the revised manuscript.
>
> Following this suggestion, we added experiments on two additional widely used pretrained models, **Llama-3.2-1B-Base and Gemma3-1B-pt**, as baselines.
> Each experiment was run on 64 H800 GPUs for 1–2 days.
> We then compared the performance of the vanilla models and their DND-enhanced counterparts across a diverse set of benchmarks.
>
> As shown in the table below, applying DND to Llama-3.2-1B and Gemma3-1B leads to clear performance gains.
> Across more than ten benchmarks—including general tasks, math, coding, agent,
> and alignment tasks—the average improvements are **2.61** and **2.50** points, respectively.
> These results indicate that the benefits of DND are not restricted to the Qwen family of models.
>
> (_Details can be referred to Sec. 4.3 and Tab.1 on page 7 of the revised manuscript._)
>
> |                 |             | **General Knowledge & Alignment** |           |            |           |            | **Math & STEM** |           | **Coding & Agent** |           |           |              |
> |:----------------|:-----------:|:---------------------------------:|:---------:|:----------:|:---------:|:----------:|:---------------:|:---------:|:------------------:|:---------:|:---------:|:------------:|
> |                 | **Average** |              **BBH**              | **PIQA**  | **C-Eval** | **MMLU**  | **IFEval** |    **GPQA**     | **GSM8K** |      **MBPP**      | **Human+** | **BFCL**  | **MultiPLE** |
> | **Llama3.2-1B** |    45.37    |               25.73               |   65.48   |   47.82    |   53.28   |   52.45    |      10.73      |   63.23   |       49.54        |   50.42   |   44.89   |    35.47     |
> | **+ DND**       |  **47.98**  |             **29.43**             | **66.51** | **49.21**  | **55.63** | **55.68**  |    **14.59**    | **66.57** |     **52.91**      | **52.16** | **47.52** |  **37.56**   |
> | *Δ (+-)*        | ***+2.61*** |              *+3.70*              |  *+1.03*  |  *+1.39*   |  *+2.35*  |  *+3.23*   |     *+3.86*     |  *+3.34*  |      *+3.37*       |  *+1.74*  |  *+2.63*  |   *+2.09*    |
> | **Gemma3-1B**   |    47.08    |               25.93               |   70.27   |   50.14    |   55.98   |   54.52    |      16.49      |   65.53   |       49.29        |   52.62   |   40.69   |    36.43     |
> | **+ DND**       |  **49.58**  |             **30.62**             | **71.33** | **51.00**  | **58.04** | **56.91**  |    **21.79**    | **68.68** |     **52.96**      | **53.76** | **42.88** |  **37.41**   |
> | *Δ (+-)*        | ***+2.50*** |              *+4.69*              |  *+1.06*  |  *+0.86*   |  *+2.06*  |  *+2.39*   |     *+5.30*     |  *+3.15*  |      *+3.67*       |  *+1.14*  |  *+2.19*  |   *+0.98*    |

---

> ### Author Response · Authors · 2025-11-26
>
> Dear Reviewer pkVU,
>
> We hope our follow-up response and supplementary experiments help to address your questions. Your review is very valuable, and we would appreciate your further feedback. Of course, please let us know if this or any other point remains unclear, as we would be happy to elaborate further.
>
> Thank you for your valuable time!
>
> Sincerely,
>
> Authors of Paper 8392

---

> ### Author Response · Authors · 2025-11-27
>
> Dear Reviewer pkVU,
>
> We deeply appreciate your thoughtful review and the time you've dedicated to our work. We truly hope that our follow-up response and supplementary experiments have addressed your concerns. Your feedback is incredibly valuable to us, and we would be truly grateful for any additional thoughts or suggestions you may have.
>
> If there are any points that remain unclear or if further elaboration is needed, please do not hesitate to reach out. We would be more than happy to provide any clarification you may need.
>
> Thank you once again for your time and consideration.
>
> Warm regards,
>
> Authors of Paper 8392

---

> ### Author Response · Authors · 2025-11-28
>
> Dear Reviewer pkVU,
>
> We sincerely thank you for your time and constructive feedback. We hope our response and supplementary experiments have adequately addressed your concerns. Please let us know if there is anything else we can elaborate on.
>
> Thank you once again for your time and consideration.
>
> Best regards,
>
> Authors of Paper 8392

---

### Author Response · Authors · 2025-11-19
**Genneral Summarization of Rebuttal**

We thank the PCs, SACs, ACs, and all reviewers for their efforts and valuable feedback.
In the first-time rebuttal, we have mainly made the following three improvements to the paper:

1. **Additional LLM experiments.** We added SFT experiments with and without DND on Llama-3.2-1B and Gemma3-1B.
Across more than ten benchmarks, DND yields average improvements of +2.61 and +2.50 points over vanilla SFT, respectively.
Together with the original results on Qwen3-1.7B and Qwen3-30B-A3B, these experiments show that our method provides stable gains
across different architectures and parameter scales.

2. **Quantitative evidence for motivation and effectiveness.**
We now include quantitative analyses showing that the DND router tends to select tokens with higher logits entropy in the vanilla forward pass,
which indicates lower model confidence and higher learning difficulty, aligning with our motivation.
We also show that, after applying DND, the entropy of the logits for these tokens decreases notably,
indicating that the model learns these harder tokens more effectively.

3. **Throughput comparison in practical inference.**
We added real-time throughput measurements comparing our DND-enhanced models with their baselines under standard settings.
The results show that our models retain 91.6–93.1% of the baseline throughput while achieving substantially better performance.

We once again thank the chairs and reviewers for their time and consideration.
We are looking forward to receiving further feedback from the reviewers and are willing to answer any additional questions.

---

### Author Response · Authors · 2025-11-25
**Concise Summary of Submission 8392**

Dear Program Chairs, Senior Area Chairs, and Area Chairs,

Firstly, we'd like to express our sincere gratitude for taking the time to review our paper.

To facilitate understanding, we summarize below our main contributions and the key improvements made during the rebuttal period:

* We introduce **Dynamic Nested Depth (DND)**, an efficient paradigm that adaptively identifies critical tokens and selectively deepens their computation through nested re-processing. We further conduct **quantitative experiments** verifying the **effectiveness of our critical-token selection mechanism** and demonstrating that **DND facilitates more confident representations for these tokens**.

* DND is the **first method to consistently improve pretrained LLM performance** through **effective vertical depth expansion**, all while keeping the parameter count unchanged and incurring only a minor reduction in throughput.

* Extensive experiments show that DND can be seamlessly integrated into both **dense LLMs** (**extra average gains beyond vanilla SFT**: Qwen3-1.7B + 1.88, Llama-3.2-1B + 2.61, Gemma3-1B + 2.50) and **MoE architectures** (scaling successfully up to **Qwen3-30B-A3B**) via post-training, yielding **consistent gains across more than ten challenging large-scale benchmarks**.

Thank you once again for your time and consideration.

Best regards,

Authors of Paper 8392

---

### Author Response · Authors · 2025-12-03

Dear Program Chairs, Senior Area Chairs, and Area Chairs,

Firstly, we would like to express our sincere gratitude for taking the time to review our paper.

In our submission, we introduce Dynamic Nested Depth (DND), an efficient framework for large language models that adaptively identifies critical tokens and deepens their computation through nested re-processing. For your convenience, we have summarized the main contributions in the section *“Concise Summary of Submission 8392”*.

All three reviewers recognized the intuitive and well-justified motivation behind our approach, as well as the reasonableness and effectiveness of our proposed solution. However, they raised concerns regarding the generalizability of the model architecture, wall-clock latency, and the lack of quantitative experiments to validate the motivation and effectiveness. In response, we have made targeted revisions to the manuscript, which are summarized in the section *“General Summarization of Rebuttal”*.

As noted in the section *“Resolving Reviewer Concerns Pending Further Feedback”*, although we believe we have thoroughly addressed all reviewer concerns, the early closure of the discussion phase prevented us from receiving timely feedback from the reviewers.

We hope the above summaries help you gain a clear understanding of our submission.

Thank you once again for your time and consideration.

Best regards,
Authors of Paper 8392

---

### Meta-Review · Area_Chair_soMR · 2026-01-07

**Summary:**

The reviewers were split about this paper and did not come to a consensus. On one hand they appreciated the model architecture and the ablations. On the other they had issues with (a) small improvements, (b) similarity to existing ideas, (c) limited architectures, (d) a missing computation-matched comparison, (e) missing details on SFT. The authors responded to all of these points convincingly. As all of the concerns have been responded to, I vote to accept.

**Reviewer Concerns:**

Please see above.

**Reviewer Scores:**

I think all reviewers would have kept their score or increased it.

---

### Decision · Program_Chairs · 2026-01-26

Accept (Poster)